# TEST-TIME GRAPH SEARCH FOR GOAL-CONDITIONED REINFORCEMENT LEARNING

## ABSTRACT

Offline goal-conditioned reinforcement learning (GCRL) trains policies that reach user-specified goals at test time, providing a simple, unsupervised, domain-agnostic way to extract diverse behaviors from unlabeled, reward-free datasets. Nonetheless, long-horizon decision making remains difficult for GCRL agents due to temporal credit assignment and error accumulation, and the offline setting amplifies these effects. To alleviate this issue, we introduce Test-Time Graph Search (TTGS), a lightweight planning wrapper for pretrained GCRL policies which only uses the pretraining dataset. TTGS accepts any state-space distance or cost signal, builds a weighted graph over dataset states, and performs fast search to assemble a sequence of subgoals that a frozen policy executes. When the base learner is value-based, the distance is derived directly from the learned goal-conditioned value function, so no handcrafted metric is needed. TTGS requires no changes to training, no additional supervision, no online interaction, and no privileged information, and it runs entirely at inference. On the OGBench benchmark, TTGS improves success rates of multiple base learners on challenging locomotion tasks, demonstrating the benefit of simple metric-guided test-time planning for offline GCRL.

## 1 INTRODUCTION

Goal-conditioned reinforcement learning trains agents to reach user-specified goals and offers a simple interface for extracting diverse behaviors from large datasets. The goal specification decouples behavior from specific task rewards, which enables reuse across many objectives and facilitates broad data curation without fragile reward engineering (Schaul et al., 2015; Andrychowicz et al., 2017). In the offline regime, GCRL can leverage precollected data in domains where online interaction is expensive, time-consuming, or unsafe, for example robotics, healthcare, and autonomous driving (Fu et al., 2020; Abdellatif et al., 2021; Kumar et al., 2020).

Despite these advantages, long-horizon decision making remains challenging in offline settings. Recent evaluations report that methods performing well in moderate-horizon environments struggle in larger spaces such as bigger mazes (Park et al., 2025; Sobal et al., 2025).

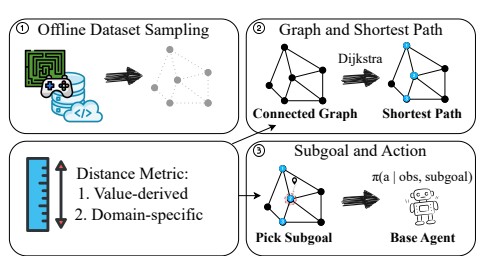

Figure 1: **Overview of TTGS.** From an offline dataset, we sample observations to form graph vertices. We assign edge weights using a distance signal, either derived from a pretrained goal-conditioned value function or from domain-specific knowledge. A shortest-path search with Dijkstra's algorithm yields a sequence of subgoals that guides a frozen policy at test time.

Effective solutions often decompose progress into reachable subproblems. For example, they learn high-level policies that output subgoals (Park et al., 2023), sample candidate trajectories with generative models (Ajay et al., 2023; Luo et al., 2025), or combine graph search with learned geometric representations and low-level controllers (Eysenbach et al., 2019; Baek et al., 2025). However, these approaches ~~still~~ usually require specialized training objectives, such as

distributional value learning (Eysenbach et al., 2019), or additional data beyond the offline pretraining dataset (Emmons et al., 2020).

~~In many cases, existing agents already have access to information about the geometry of the environment, either explicitly through domain knowledge such as kinematic feasibility, terrain difficulty, or Euclidean proximity, or implicitly through a~~ In contrast, we demonstrate that standard goal-conditioned ~~value function trained alongside the policy that encodes relative reachabilitybetween states in the dataset. If this signal can be efficiently queried at test time, it~~ value functions already contain sufficient geometric structure to support graph search, provided they are processed correctly at inference time. While these value functions can be noisy and optimistic, they implicitly encode relative reachability. By robustly querying this signal using a soft-penalty mechanism, we can guide search over dataset states ~~and~~ to yield a path where each segment remains within the capabilities of the pretrained policy~~.~~, without the need for specialized training or ensembles. We further show that this framework generalizes to domain-specific heuristics, such as Euclidean proximity.

We introduce *Test-Time Graph Search* (TTGS), a lightweight planning wrapper for pretrained GCRL policies. TTGS constructs a weighted graph over dataset observations, assigns edge costs using a distance function, and performs a fast shortest-path search to return a sequence of intermediate subgoals that the policy executes. Our main instantiation uses a value-derived distance obtained directly from the learned goal-conditioned value function, which removes the need for hand-crafted metrics. We also study a domain-specific instantiation based on agent body position to illustrate that the framework can accommodate task-informed metrics when available. In both cases, planning over the induced geometry significantly improves trajectory stitching and long-horizon reasoning without modifying the underlying policy or requiring retraining, additional supervision or online interaction.

TTGS integrates with existing goal-conditioned learners as an inference-time procedure. It requires only a distance signal and an offline dataset, adds minimal computational overhead, and preserves the behavior learned by the base policy. The framework complements both hierarchical and non-hierarchical offline GCRL methods by supplying explicit long-horizon search at test time. Since many GCRL algorithms already learn goal-conditioned value functions, the value-derived distance is a natural default that keeps the method simple to adopt.

We make the following contributions:

- We ~~identify a specific cause of long-horizon failures in value-based GCRL: policies fail to exploit the geometric structure encoded in their~~ demonstrate that standard goal-conditioned value functions ~~. By using the value function to compute distances and planning subgoalsover dataset states~~ contain sufficient geometric structure for planning, provided they are processed with a soft-penalty mechanism to suppress noise. By leveraging this derived geometry to plan subgoals, we show that the same policies can achieve substantially better performance without any additional training.

- We present TTGS, a test-time planning framework that builds a graph over dataset states and uses shortest-path search to produce subgoal sequences. The framework supports value-derived distances and also admits domain-specific distances when available.

- We demonstrate consistent gains on OGBench long-horizon locomotion tasks with both value-derived and domain-specific distances, improving over base learners and sometimes even surpassing substantially more complex methods that require extra training (Park et al., 2025; Luo et al., 2025; Baek et al., 2025).

## 2 GOAL-CONDITIONED REINFORCEMENT LEARNING PRELIMINARIES

Following Park et al. (2025), we consider the offline goal-conditioned reinforcement learning (GCRL) problem, defined over a controlled Markov process $\mathcal{M} = (\mathcal{S}, \mathcal{A}, \mu, p)$, i.e, a Markov decision process (MDP) without rewards, together with an unlabeled dataset $\mathcal{D}$ of trajectories. Here, $\mathcal{S}$ is the state space, $\mathcal{A}$ the action space, $\mu \in \Delta(\mathcal{S})$ the initial state distribution, and $p(s' \mid s, a) : \mathcal{S} \times \mathcal{A} \to \Delta(\mathcal{S})$ the transition dynamics. We denote by $\Delta(\mathcal{X})$ the space of probability distributions over a set $\mathcal{X}$.

The offline dataset $\mathcal{D} = \{\tau^{(n)}\}_{n=1}^N$ consists of trajectories $\tau^{(n)} = (s_0^{(n)}, a_0^{(n)}, s_1^{(n)}, \ldots, s_{T_n}^{(n)})$ collected by some unknown behavior policy.

The goal in GCRL is to learn a goal-conditioned policy $\pi(a \mid s, g): \mathcal{S} \times \mathcal{S} \to \Delta(\mathcal{A})$ that can efficiently reach any goal $g \in \mathcal{S}$ from any starting state $s \in \mathcal{S}$. Formally, the optimal policy maximizes the expected discounted return:

$$\max_{\pi} \; \mathbb{E}_{\tau \sim p(\tau|g)} \left[ \sum_{t=0}^{T} \gamma^t \, \mathbf{1}\{\|s - g\| < \epsilon\} \right],$$

where $T$ is the horizon, $\gamma \in (0, 1)$ the discount factor, $p(\tau \mid g)$ the trajectory distribution induced by $\pi$, and reward is equal to $\mathbf{1}\{\|s - g\| < \epsilon\}$.

Many GCRL algorithms, including HIQL (Park et al., 2023), GCIQL (Kostrikov et al., 2022), and QRL (Wang et al., 2023), learn a value function $V(s, g): \mathcal{S} \times \mathcal{S} \to \mathbb{R}$ representing the expected discounted reward when navigating from state $s$ to goal $g$.

## 3 METHOD

We visualize the core challenge that GCRL agents face in Figure 2. In long, complex tasks like maze navigation, agents that attempt to reach far-away goals can get stuck or run off course. However, for shorter horizons, they tend to be reliable. From this observation, we derive the simple key idea behind TTGS: rather than asking a policy to solve a long-horizon task in one shot, we decompose it

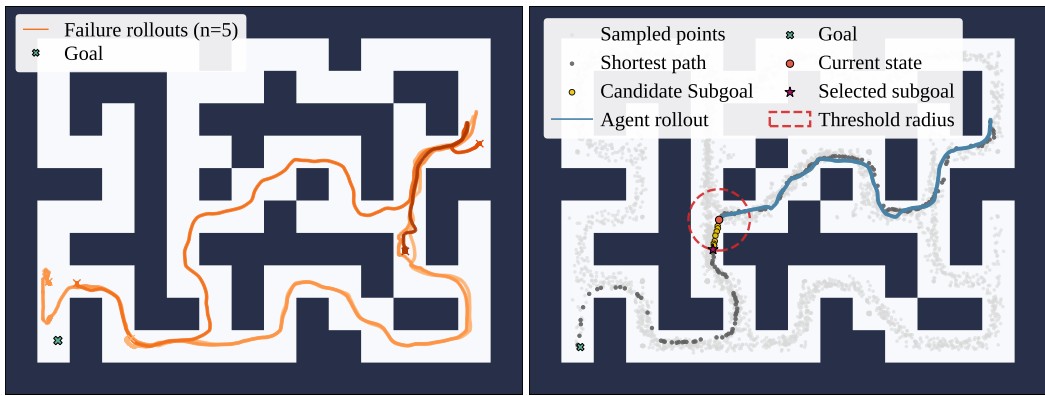

(a) Rollouts from HIQL fail to reach a distant goal.    (b) TTGS selects reachable subgoals on a guiding path.

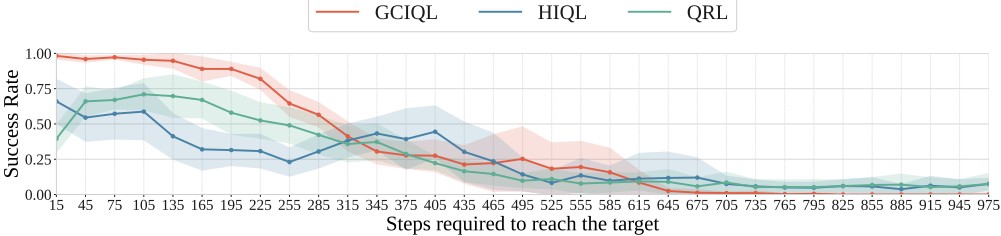

(c) Success rate of reaching goals located $n$ steps away from the agent within $1.5n$ step budget in `antmaze-giant-stitch-v0` environment.

Figure 2: **Motivation for TTGS: (a)** HIQL policy fails to reach a distant goal on `antmaze-giant-stitch-v0`, with multiple attempts failing to exit the starting area and two attempts running out of time due to inefficient path. **(b)** TTGS finds a guiding path using dataset observations. On each step it selects a subgoal which is within a predefined radius from the agent. We mark all data points on the guiding path in gray, and the actual path traversed by the agent in blue. **(c)** Different agents' policy performance decreases as steps required to reach the goal increase. By providing a policy with close subgoals, TTGS improves reliability and efficiency of reaching the goal.

into short, reliable hops. We do so by equipping test-time planning with a distance signal over states, selecting a compact set of representative states from an offline dataset, and connecting them into a graph whose edges reflect predicted step costs. At test time, we plan in this graph by computing a shortest path between the start and the goal, then feed the policy a small number of intermediate subgoals along this path. As summarized in Figure 2b, the agent queries all points along the shortest path that are within a certain threshold from the current state, and picks the closest to the goal.

While the framework supports any compatible distance, our primary focus is the general and widely applicable case where the distance is derived from a learned goal-conditioned value function. The following sections describe how distances are calibrated (Section 3.1), how the graph is constructed (Section 3.2), and how subgoal-based planning is executed at test time (Section 3.3).

### 3.1 DISTANCE PREDICTION

The algorithm begins by obtaining a distance predictor from an available (approximate) metric on the state space. In general, this can be any metric, here we focus on deriving approximate metrics from value function estimates. Note that in general these predictors do not fulfill the formal criteria for metrics, e.g. they might not obey the triangle inequality or be symmetric, but in our experiments. However, we find them to be very reliable at least for nearby goal states reliable for local reachability (i.e., within a short horizon), even if global long-horizon estimates are noisy, as supported by the effectiveness of TTGS in chaining these local estimates.

**Value-derived distance (default).** To enable policy-agnostic planning, we map a goal-conditioned value signal $V(s, g)$ to an expected step distance

$$\hat{d}(s, g) = f\big(V(s, g)\big),$$

where $f : \mathbb{R} \to \mathbb{R}_{\geq 0}$ converts values into an estimate of the environment steps required to reach $g$ from $s$. For the following, common reward conventions, $f$ admits a closed form:

- *Sparse terminal reward.* Reward $1$ only at the goal and $0$ otherwise. Then $V^*(s, g) = \gamma^d$, where $d$ is the shortest-step distance, giving

$$\hat{d}(s, g) = \log_\gamma V^*(s, g).$$

- *Per-step penalty.* Reward $-1$ until reaching the goal and $0$ at the goal. Then

$$V^*(s, g) = -\sum_{t=0}^{d-1} \gamma^t = -\frac{1 - \gamma^d}{1 - \gamma} \quad \Rightarrow \quad \hat{d}(s, g) = \log_\gamma\big(1 + (1 - \gamma)\, V^*(s, g)\big).$$

Both HIQL HIQL, SAW and GCIQL use this convention in our experiments.

More generally, any value signal that is monotone with respect to reachability suffices. We clip $\hat{d}$ to avoid negative or infinite outputs. Exact mappings for each base agent are provided in Appendix B.

**Domain-specific distances.** When a task supplies a meaningful geometric or kinematic metric, we can use it directly as $\hat{d}$. This requires no changes to the pipeline below; only the source of edge costs differs. Our experiments include such a domain-specific metric to illustrate the flexibility of the framework.

### 3.2 GRAPH CONSTRUCTION

Using the distance predictor, the agent now proceeds to build the underlying graph of distances from the pretraining dataset. This graph is goal-agnostic, which means it can be reused for different goals an agent might want to achieve in the environment.

**Vertices selection** Computing all pairwise distances among $N$ states in $\mathcal{D}$ has $O(N^2)$ complexity and may require repeated network evaluations, which is prohibitive at scale. We instead sample a compact subset $\mathcal{V} = \{v_i\}_{i=1}^M \subset \mathcal{D}$ and build the graph on $\mathcal{V}$. Uniform random sampling worked well in our experiments. We also tested a trajectory efficiency filter and clustering inspired by Baek et al. (2025) (see Appendix D for details), but it did not produce a statistically significant gain over random sampling, so we adopt random sampling for simplicity.

---

**Algorithm 1** TTGS: Subgoal Selection and Execution

---

**Require:** Graph $G = (\mathcal{V}, E, \tilde{w})$ where $\mathcal{V} = \{v_i\}_{i=1}^M$ are selected states; distance $\hat{d}$; frozen policy $\pi$; start state $s_0$; goal $g$; step budget $T > 0$

1: **// Precompute guide path once per episode**
2: $v_s \leftarrow \arg\min_{v_i} \hat{d}(s_0, v_i), \quad v_g \leftarrow \arg\min_{v_i} \hat{d}(v_i, g)$
3: $(p_0, \ldots, p_L) \leftarrow \text{DIJKSTRA}(G, v_s, v_g)$
4: $k_{\text{prev}} \leftarrow 0; \quad s_{\text{cur}} \leftarrow s_0$
5: **while** $g$ not reached **do**                                    // main control loop
6: $\quad \delta_\ell \leftarrow \hat{d}(s_{\text{cur}}, p_\ell)$ for $\ell = 0, \ldots, L; \quad \delta_g \leftarrow \hat{d}(s_{\text{cur}}, g)$
7: $\quad k \leftarrow \arg\min_\ell \delta_\ell; \quad k \leftarrow \max(k, k_{\text{prev}})$     // index of closest waypoint ahead
8: $\quad$ **if** $\delta_g \leq T$ **then**                               // goal is within reach
9: $\quad\quad \tilde{g} \leftarrow g$
10: $\quad$ **else**
11: $\quad\quad \mathcal{C} \leftarrow \{\ell > k : \delta_\ell \leq T\}$          // reachable subgoal candidates
12: $\quad\quad$ **if** $\mathcal{C} \neq \varnothing$ **then**
13: $\quad\quad\quad \tilde{g} \leftarrow p_{\max \mathcal{C}}$                    // take farthest reachable
14: $\quad\quad$ **else**
15: $\quad\quad\quad \tilde{g} \leftarrow p_{\min\{k+1, L\}}$                 // otherwise take next subgoal
16: $\quad\quad$ **end if**
17: $\quad$ **end if**
18: $\quad$ Sample $a \sim \pi(\cdot \mid s_{\text{cur}}, \tilde{g})$ and execute     // act with frozen policy toward subgoal
19: $\quad s_{\text{cur}} \leftarrow$ next observed state; $\quad k_{\text{prev}} \leftarrow k$
20: **end while**

---

**Weights derivation**   We build a directed, weighted graph $G = (\mathcal{V}, E, \tilde{w})$. Using the distance predictor $\hat{d}$, we compute a matrix $D \in \mathbb{R}^{M \times M}$ whose entries are $D_{ij} = \hat{d}(v_i, v_j)$, using batched evaluations for efficiency. However, using $D$ directly for planning can produce paths that jump across large gaps. When the triangle inequality holds, inserting intermediate vertices cannot reduce cost, so the planner prefers single long hops. In addition, value-derived distances can be optimistic and may connect states that are hard or even impossible to traverse. To shape the geometry of the resulting routes, we apply a soft horizon that trusts distances only up to a threshold $\tau > 0$. Edges with $D_{ij} < \tau$ keep their predicted costs, edges with $D_{ij} \geq \tau$ receive a superlinear penalty $p$, and self-loops are removed by setting their weight to $+\infty$:

$$
\tilde{w}_{ij} = \begin{cases} D_{ij}, & i \neq j, \ D_{ij} < \tau, \\ p(D_{ij}), & i \neq j, \ D_{ij} \geq \tau, \\ +\infty, & i = j. \end{cases}
$$

In experiments we use $p(x) = x \cdot 1000^{x/\tau}$. Leaving costs unchanged below $\tau$ preserves local metric structure and keeps TTGS optimal with respect to $\hat{d}$ within trusted neighborhoods when coverage is dense. Penalizing long edges suppresses unreliable shortcuts without disconnecting the graph and steers shortest paths toward sequences of short, reliably reachable hops. Finally, as our graph-traversal algorithm only selects goals within a short threshold radius of the agent's current position, we do not drastically change the final path with these changes.

### 3.3 SUBGOAL SELECTION AND ACTION SAMPLING

To find the path to a given goal, the agent proceeds in two steps based on the pre-computed graph.

**Shortest-path precomputation.**   At test time, given a start state $s_0$ and a goal $g$, we compute a guide path on $G$ using Dijkstra's algorithm. We first locate the nearest vertices under $\hat{d}$,

$$
v_s = \arg\min_{v_i} \hat{d}(s_0, v_i), \qquad v_g = \arg\min_{v_i} \hat{d}(v_i, g),
$$

then obtain $(p_0, \ldots, p_L) = \text{DIJKSTRA}(G, v_s, v_g)$. The guide path is computed once per episode and reused throughout execution. Using a fast GPU-based implementation of Dijkstra's algorithm keeps the overhead minimal.

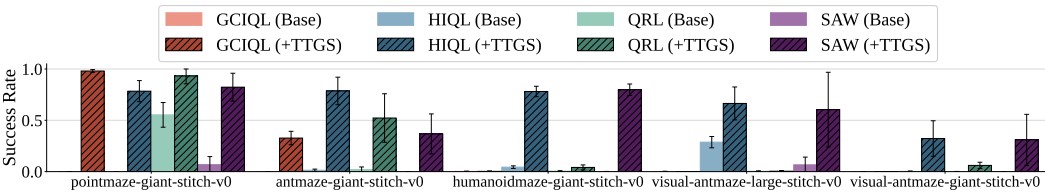

Figure 3: ~~Goal-reaching success rates for QRL, GCIQL, and HIQL with and without TTGS.~~ **Goal-reaching success rates for QRL, GCIQL, HIQL and SAW with and without TTGS.** Distances are predicted from each base agent's learned value function. TTGS consistently improves or preserves performance on locomotion tasks that require trajectory stitching.

**Adaptive subgoal selection.** During execution, the agent selects subgoals from $(p_0, \ldots, p_L)$ using a step budget $T$. If the current state $s_{\text{cur}}$ is within $T$ steps of $g$ according to $\hat{d}$, the subgoal is set to $g$. Otherwise, the agent chooses the farthest waypoint $p_j$ that is ahead of the closest waypoint and still within budget, that is $\hat{d}(s_{\text{cur}}, p_j) < T$. If no such waypoint exists, the next waypoint along the path is chosen to ensure forward progress. The frozen goal-conditioned policy $\pi$ is then invoked with current state and selected subgoal. The pseudocode is provided in Algorithm 1.

## 4 EXPERIMENTS

We evaluate TTGS on OGBench (Park et al., 2025) using three strong offline GCRL baselines ~~from~~ provided by the benchmark: QRL, GCIQL, and HIQL (Wang et al., 2023; Park et al., 2023). Additionally, we include SAW (Zhou & Kao, 2025) to broaden the evaluation. For each dataset, we compare the success rate of the frozen base policy with and without TTGS ~~. Each dataset contains~~ across five tasks. Every task is evaluated with 50 rollouts, and we report the mean and standard deviation over eight random seeds. In tables, methods within 95% of the best mean are bolded, and cases where TTGS exceeds its corresponding base learner are underlined. All base agents are trained with the public OGBench code using default hyperparameters, and TTGS is applied post hoc without any changes to training.

**Datasets** OGBench provides offline datasets for GCRL, including locomotion domains that test long-horizon and hierarchical reasoning (Park et al., 2025). We use `pointmaze`, `antmaze`, and `humanoidmaze` with `medium`, `large`, and `giant` mazes and three data-collection regimes: `navigate` uses a noisy expert reaching random goals, `stitch` consists of short trajectories that require trajectory stitching, and `explore` contains random exploratory rollouts (available only for `antmaze`). For `antmaze` and `humanoidmaze`, `visual` variants with pixel-based observations are also available.

### 4.1 MAIN RESULTS

Figure 3 summarizes results on the largest stitching tasks, which are challenging for existing agents. TTGS uses distances derived from the base agent's value function, so it relies only on information available at test time. TTGS generally improves or maintains success across all evaluated environments, with the largest gains on `giant` layouts where one-shot execution is difficult. On `pointmaze-giant-stitch-v0`, TTGS raises HIQL from 0.0% to 80.9% and GCIQL from 0.0% to 98.0%. On `humanoidmaze-giant-stitch-v0`, HIQL is improved from 4.4% to 78.1%. These cases highlight how sequences of short, reliable hops help frozen policies traverse large mazes that require stitching behaviors from disparate parts of the dataset. Notably, TTGS boosts even HIQL, which already includes a learned high-level controller, suggesting that explicit planning can provide more reliable subgoals than a purely learned subgoal policy in long-horizon settings. We further observe improvements on several pixel-based tasks, indicating that value-derived distances can offer a strong planning signal even when domain-specific metrics are difficult to define. ~~Full results across additional locomotion environmentsand dataset types (~~ While Figure 3 highlights the hardest tasks, we provide comprehensive results for all evaluated environments, including `medium`, `large`, and `giant` mazes across `stitch`, `navigate`, and `explore` ~~) as well as comparisons~~

Table 1: **HIQL+TTGS on state-based datasets using value-derived and position-based ($L_2$) distances**. The simple Euclidean distance between body positions, normalized by the average dataset step length, is often sufficient to yield strong gains.

| Dataset | HIQL | HIQL+TTGS-value | HIQL+TTGS-$L_2$ |
|---|---|---|---|
| `pointmaze-giant-navigate-v0` | $43.0 \pm 10.5$ | $\mathbf{70.9 \pm 12.2}$ | $\mathbf{70.0 \pm 9.3}$ |
| `pointmaze-giant-stitch-v0` | $0.0 \pm 0.0$ | $\mathbf{80.9 \pm 9.0}$ | $\mathbf{83.0 \pm 7.8}$ |
| `antmaze-giant-navigate-v0` | $\mathbf{65.0 \pm 4.1}$ | $65.8 \pm 4.0$ | $60.8 \pm 4.3$ |
| `antmaze-giant-stitch-v0` | $1.4 \pm 1.1$ | $\mathbf{78.6 \pm 13.4}$ | $60.2 \pm 12.0$ |
| `antmaze-large-explore-v0` | $2.4 \pm 4.4$ | $\mathbf{26.6 \pm 34.0}$ | $18.8 \pm 23.8$ |
| `humanoidmaze-giant-navigate-v0` | $16.0 \pm 8.6$ | $\mathbf{85.3 \pm 6.1}$ | $74.2 \pm 17.5$ |
| `humanoidmaze-giant-stitch-v0` | $4.4 \pm 1.3$ | $\mathbf{78.1 \pm 5.1}$ | $72.6 \pm 5.7$ |

~~to other planning methods are provided in~~ settings, in Table 3 ~~of~~ Appendix A. We further analyze the applicability of TTGS to manipulation tasks in Appendix C, finding that the method behaves robustly even when value landscapes are sparse.

To test the generality of TTGS beyond value-based distances, we also evaluate a simple domain-specific distance based on body position. This metric uses the Euclidean distance between agent body positions and normalizes by the dataset's average step length. Although it ignores joint configuration and contact dynamics, this signal is sufficient to unlock substantial gains when combined with TTGS. Table 1 reports that HIQL+TTGS with this metric improves performance across most state-based environments, which supports the claim that TTGS can exploit a variety of compatible distances when a value function is unavailable.

### 4.2 ABLATION STUDY

We study two design choices: adaptive subgoal selection and the edge-length penalty introduced during graph construction. We also examine sensitivity to the distance penalty threshold $\tau$ and the subgoal selection threshold $T$.

**Subgoal selection.** Our default procedure selects the farthest waypoint on the precomputed route whose predicted distance from the current state is below a threshold $T$, falling back to the next

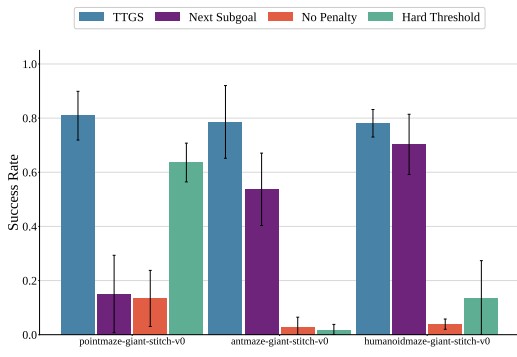
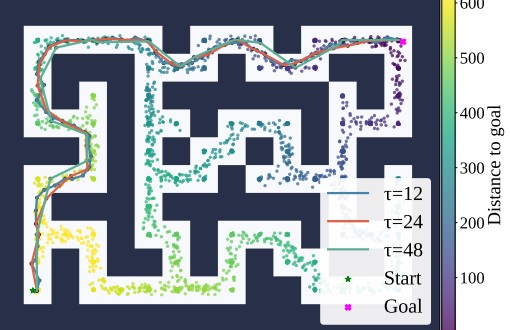

(a) Ablations of HIQL+TTGS-value.  (b) Effect of the threshold $\tau$ and predicted distances.

Figure 4: **Ablations and hyperparameters. (a)** Comparison of full TTGS using HIQL as base learner and value-derived distances with ~~two~~ three ablations: *Next Subgoal* replaces our subgoal selection procedure with always picking the immediate next waypoint, ~~and~~ *No-Penalty* uses raw predicted distances~~as edge weights instead of penalizing long connections~~, and *Hard-Threshold* removes edges with cost $> \tau$. TTGS outperforms both ablations across datasets. **(b)** Effect of penalty threshold $\tau$ on the guide path and a value-derived distance field. Colors denote predicted distances from each dataset observation to the goal in top-right corner. Smaller $\tau$ yields denser subgoals and less direct paths. Larger $\tau$ permits longer hops that can require navigating around obstacles, which is harder for the frozen policy.

Table 2: **TTGS consistently boosts HIQL on long-horizon stitch tasks.** Success rate for HIQL with and without the TTGS on `pointmaze-giant-stitch-v0`, `antmaze-giant-stitch-v0`, and `humanoidmaze-giant-stitch-v0` as we vary $\tau$ and $T$.

| Dataset | $\tau$ | $T$ | HIQL | HIQL+TTGS |
|---|---|---|---|---|
| | 12 | 24 | | **78.4 ± 10.3** |
| | 24 | 24 | | 57.2 ± 12.7 |
| `pointmaze-giant-stitch-v0` | 24 | 48 | 0.0 ± 0.0 | **80.9 ± 9.0** |
| | 24 | 96 | | 62.6 ± 16.7 |
| | 48 | 96 | | 52.0 ± 20.0 |
| | 12 | 24 | | **78.6 ± 13.4** |
| | 24 | 24 | | 40.9 ± 14.6 |
| `antmaze-giant-stitch-v0` | 24 | 48 | 1.4 ± 1.1 | 19.0 ± 9.2 |
| | 24 | 96 | | 2.2 ± 1.6 |
| | 48 | 96 | | 1.1 ± 1.5 |
| | 12 | 24 | | 52.2 ± 6.1 |
| | 24 | 24 | | 65.5 ± 12.0 |
| `humanoidmaze-giant-stitch-v0` | 24 | 48 | 4.4 ± 1.3 | **78.1 ± 5.1** |
| | 24 | 96 | | **79.8 ± 7.7** |
| | 48 | 96 | | **76.8 ± 9.7** |

waypoint after the closest node when no such waypoint exists. The alternative always commits to the immediate next waypoint. Figure 4a shows that this simpler strategy degrades performance, especially on `pointmaze-giant`, while the adaptive rule yields robust improvements. We hypothesize that aiming at the farthest reachable waypoint reduces unnecessary micromanagement of local motion and promotes efficient progress toward reachable, yet sufficiently distant, subgoals.

**Distance penalty.** We compare the dynamic soft penalty from Section 3.2 to a *No-Penalty* variant that uses the predicted distances directly as edge weights. Penalizing long edges consistently improves success by discouraging unreliable shortcuts while keeping local geometry intact(raw distances) and a *Hard-Threshold* variant (removing edges $> \tau$). Figure 4a shows that on stitching tasks, the soft penalty significantly outperforms the no-penalty baseline. Furthermore, as detailed in Appendix E, a hard threshold fails on these tasks because noisy value estimates frequently cause graph disconnections. The soft penalty maintains connectivity by assigning high costs to optimistic edges rather than removing them, allowing the planner to recover from local value function errors.

**Hyperparameters.** We sweep the edge threshold $\tau$ and the subgoal threshold $T$. Table 2 reports HIQL results on three biggest state-based mazes. TTGS outperforms the base policy or preserves the performance across all tested settings, with a moderate penalty threshold and a subgoal selection threshold near $T = 2\tau$ generally working best. Small $\tau$ values bias the planner toward long chains of short hops, which can lengthen routes and reduce straight-line progress, whereas large $\tau$ values admit long edges that the policy may fail to traverse. In practice, choosing hyperparameters for TTGS is simple, since the method does not require training which allows for fast iteration.

**Replanning.** We also investigate the effect of online replanning in Appendix G. We find that the one-time guide path is generally robust, and frequent replanning typically offers limited benefit.

Finally, Figure 4b visualizes the value-derived distance field together with guide paths produced by TTGS. The distances obtained from the value function reflect the environment's structure, with obstacles forming high-cost barriers and corridors forming low-cost channels. As the edge threshold $\tau$ decreases, the planner trusts only short edges and yields denser routes with more subgoals; as $\tau$ increases, longer edges are permitted, producing sparser but potentially less reliable paths.

## 5 RELATED WORK

Offline goal-conditioned reinforcement learning (GCRL) aims to learn multi-task policies from fixed datasets. A key challenge is to infer reliable goal-conditioned values from imperfect data. Prior

approaches stabilize value estimation with expectile regression (Kostrikov et al., 2022; Park et al., 2023), treat values as quasi-metric distances (Wang et al., 2023), or couple representation learning with one-step improvement via contrastive objectives (Eysenbach et al., 2022; Zheng et al., 2024). These methods perform well on moderate horizons but often fail on tasks that require trajectory stitching. TTGS is designed to complement them by supplying test-time planning to improve their long-horizon performance.

Graph-based methods have also been explored in GCRL. Some use graph search to prune suboptimal actions from datasets (Yin & Abbeel, 2024), while others construct graphs from episodes in a learned metric space and apply reinforcement learning over the graph (Zhu et al., 2023). Several works focus on learning more accurate distance estimators, including temporal distance representations (Baek et al., 2025), ensembles of distributional value functions (Eysenbach et al., 2019), Siamese proximity predictors (Savinov et al., 2018), or latent landmarks (Zhang et al., 2021). Some approaches estimate distances via value functions but require online refinement to prune infeasible edges (Emmons et al., 2020). Unlike prior graph search approaches, TTGS requires no training, it treats pretrained policies and distance estimators as black boxes and operates entirely offline. We show that instead of using specialized training or online pruning, applying a simple soft threshold and subgoal selection on top of standard pretrained agents is sufficient to gain the benefits of graph search.

Beyond graph search, several planning paradigms have been proposed. Diffusion and generative models compose or augment trajectories for long-horizon reasoning (Janner et al., 2022; Ajay et al., 2023; Luo et al., 2025; Zhang et al., 2025; Lee & Choi, 2025), though they typically demand extra training and computation. Model-based planning uses learned dynamics for model-predictive control (Williams et al., 2017; Chua et al., 2018; Hafner et al., 2019; Hansen et al., 2022) or sequence-model planning with trajectory transformers (Janner et al., 2021). TTGS differs by providing a lightweight, training-free planning layer over dataset states, which can complement generative and model-based methods by providing reliable subgoals.

## 6 LIMITATIONS

TTGS introduces some computational overhead, although it remains modest. Graph construction is performed once per dataset and requires roughly ~~45~~ 35 to 100 seconds on a single ~~L40 GPU for state-based environments and about 100 seconds for visual environments~~ Nvidia L40s GPU with 8 virtual CPU cores, depending on the domain. At test time, computing a shortest path is required once per episode and ~~takes about 0.8 seconds~~ typically takes less than one second, which is competitive with existing planning methods. During execution, action selection adds ~~only a ≈ 1~~ ≈ 1.5ms overhead per step. We provide a detailed runtime breakdown in Appendix F. In practice, these costs are minor compared to the performance gains, and the absence of additional training allows practitioners to iterate quickly.

The effectiveness of TTGS depends on the base learner. The framework assumes that the policy can reliably reach nearby subgoals, and its performance is influenced by both the accuracy of the distance estimator and the coverage of states in the offline dataset. In some environments we observe substantial improvements from near-zero baselines, but in others the base policies are too unreliable to follow subgoal chains effectively. Similarly, when goal-conditioned value functions provide noisy or inconsistent distance estimates, TTGS may struggle to produce useful guidance, although in such cases the base learners themselves already perform poorly.

Finally, although we do not observe statistically significant degradations compared to base learners, TTGS may select subgoals that are suboptimal when dataset coverage is limited or distance predictions are inaccurate. Future work could explore combining multiple noisy distance estimates to obtain more robust edge weights, as well as developing more sophisticated strategies for selecting representative states as graph vertices.

## 7 CONCLUSION

We introduced Test-Time Graph Search (TTGS), a lightweight planning framework that augments pretrained goal-conditioned reinforcement learning agents. TTGS leverages only the offline dataset and a distance signal to construct a graph, then applies efficient search to generate subgoal sequences

for a frozen policy. This approach requires no retraining, no additional supervision, and no online interaction, yet substantially improves long-horizon performance across diverse tasks.

Our study highlights a key reason why value-based GCRL agents struggle on long-horizon problems: their policies fail to exploit the geometric structure already captured by goal-conditioned value functions. By reinterpreting these value functions as distances and planning over dataset states, TTGS shows that the same policies can achieve significantly stronger performance.

Experiments on OGBench confirm that TTGS consistently enhances the reliability of existing agents, enabling them to solve challenging long-horizon problems that are otherwise difficult with one-shot execution. The method is effective with both value-derived distances and domain-specific metrics, underscoring its flexibility and broad applicability.

By turning offline datasets into a substrate for planning at inference time, TTGS provides a practical way to extend the capabilities of goal-conditioned learners without modifying their training pipelines. We hope this perspective inspires further exploration of simple, modular planning strategies that complement learning-based agents and increase their reliability and long-horizon performance.

## REPRODUCIBILITY STATEMENT

We provide a detailed description of the algorithm in Algorithm 1. All hyperparameters are listed in Appendix H. The code for TTGS is provided in supplementary material and will be open-sourced upon acceptance.

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

## A  FULL RESULTS

**Scope.**   We present results for TTGS paired with HIQL (Park et al., 2023), GCIQL (Kostrikov et al., 2022), ~~and~~ QRL (Wang et al., 2023), and SAW (Zhou & Kao, 2025) in Table 3. We compare against each base learner and two planning methods: GAS (Baek et al., 2025), which is a **concurrent work**, and CompDiffuser (CD) (Luo et al., 2025). Our evaluation uses a more restricted setting than both GAS and CompDiffuser. We use value-derived distances for this comparison. Results are summarized in Table 3.

**Setting differences.**   TTGS operates in a *more constrained* setting than both GAS and CompDiffuser: we perform no additional training and use only the offline pretraining dataset along with the pretrained goal-conditioned value function and frozen policy of each base learner. By contrast, GAS trains a Temporal Distance Representation (TDR) and leverages it to train a low-level controller with subgoal supervision, while CompDiffuser trains a diffusion model to synthesize long-horizon trajectories. These differences mean TTGS adds purely test-time computation, whereas GAS and CD introduce extra learned components, while also requiring planning at test time.

Table 3: **Success rates (%) across datasets.** We report base learners, their TTGS-augmented counterparts, and other planning agents where available. "/" indicates the metric was not reported in the corresponding paper. Means and standard deviations follow the OGBench protocol (50 rollouts per task; averages and s.d. over 8 seeds).

| Dataset | HIQL | HIQL +TTGS | GCIQL | GCIQL +TTGS | QRL | QRL +TTGS | SAW | SAW +TTGS | GAS | CD |
|---|---|---|---|---|---|---|---|---|---|---|
| pointmaze-medium-navigate-v0 | 73.6 ± 4.4 | 85.8 ± 5.2 | 51.5 ± 8.2 | 90.9 ± 3.7 | 83.5 ± 3.2 | **98.4 ± 3.4** | **96.8 ± 1.8** | **96.8 ± 1.9** | / | / |
| pointmaze-medium-stitch-v0 | 73.0 ± 10.2 | 80.5 ± 20.3 | 18.2 ± 8.9 | 44.0 ± 8.0 | 76.0 ± 8.9 | 93.4 ± 6.8 | 68.2 ± 7.6 | 85.4 ± 6.1 | / | **100 ± 0** |
| pointmaze-large-navigate-v0 | 45.2 ± 14.0 | 78.0 ± 9.7 | 32.2 ± 5.8 | **88.1 ± 9.4** | 82.4 ± 5.7 | **92.0 ± 9.7** | 78.0 ± 10.8 | 83.6 ± 6.4 | / | / |
| pointmaze-large-stitch-v0 | 13.2 ± 8.0 | 92.2 ± 5.1 | 29.0 ± 5.0 | 24.8 ± 4.1 | 88.8 ± 14.0 | 93.9 ± 8.7 | 41.2 ± 6.7 | 93.2 ± 6.1 | / | **100 ± 0** |
| pointmaze-giant-navigate-v0 | 43.0 ± 10.5 | 70.9 ± 12.2 | 0.0 ± 0.0 | **91.9 ± 3.0** | 65.3 ± 10.2 | ~~88.1 ± 9.5~~ 88.1 ± 9.5 | 68.5 ± 6.4 | **94.8 ± 2.8** | 5.2 ± 11.3 | / |
| pointmaze-giant-stitch-v0 | 0.0 ± 0.0 | 80.9 ± 9.0 | 0.0 ± 0.0 | **98.0 ± 1.4** | 55.3 ± 12.0 | **93.2 ± 7.7** | 6.8 ± 7.9 | 82.3 ± 13.6 | 0.0 ± 0.0 | 68 ± 3 |
| antmaze-medium-navigate-v0 | **95.2 ± 0.7** | **95.2 ± 1.3** | 72.3 ± 4.7 | 81.1 ± 4.7 | 81.9 ± 10.6 | 83.9 ± 7.8 | **96.3 ± 1.6** | 95.7 ± 1.7 | 96.3 ± 1.3 | / |
| antmaze-medium-stitch-v0 | 92.9 ± 2.1 | **95.4 ± 1.3** | 29.5 ± 4.7 | 53.0 ± 8.7 | 62.0 ± 9.4 | 40.1 ± 9.4 | 64.0 ± 5.0 | 94.0 ± 1.9 | 98.1 ± 1.2 | 96 ± 2 |
| antmaze-large-navigate-v0 | 90.6 ± 2.5 | **92.3 ± 2.3** | 35.8 ± 2.7 | 57.2 ± 3.8 | 74.0 ± 4.3 | 77.0 ± 3.3 | 88.8 ± 2.5 | **89.6 ± 1.7** | 93.2 ± 0.5 | / |
| antmaze-large-stitch-v0 | 73.0 ± 6.0 | 90.8 ± 2.3 | 7.0 ± 2.5 | 30.6 ± 4.6 | 21.0 ± 4.1 | 43.5 ± 10.5 | 3.1 ± 4.8 | 86.2 ± 3.7 | **96.3 ± 0.9** | 86 ± 2 |
| antmaze-giant-navigate-v0 | 65.0 ± 4.1 | 65.8 ± 4.0 | 0.4 ± 0.2 | 5.4 ± 2.4 | 11.8 ± 5.2 | 10.8 ± 4.8 | 68.5 ± 3.0 | 71.0 ± 5.3 | 76.0 ± 5.9 | / |
| antmaze-giant-stitch-v0 | 1.4 ± 1.1 | 78.6 ± 13.4 | 0.0 ± 0.0 | 32.7 ± 6.6 | 2.0 ± 2.6 | 52.2 ± 23.6 | 0.0 ± 0.0 | 36.8 ± 19.5 | 86.2 ± 3.6 | 65 ± 3 |
| antmaze-large-explore-v0 | 2.4 ± 4.4 | 26.6 ± 34.0 | 0.2 ± 0.5 | 66.7 ± 8.3 | 0.0 ± 0.1 | 0.0 ± 0.0 | 1.9 ± 1.8 | **91.8 ± 4.6** | 91.0 ± 9.4 | 27 ± 1 |
| humanoidmaze-medium-navigate-v0 | 88.5 ± 3.0 | **93.3 ± 2.9** | 31.8 ± 3.8 | 50.5 ± 5.5 | 19.3 ± 8.1 | 19.7 ± 8.2 | 87.9 ± 2.9 | 86.9 ± 3.1 | / | / |
| humanoidmaze-medium-stitch-v0 | 86.1 ± 3.0 | 84.6 ± 2.5 | 14.0 ± 3.6 | 20.2 ± 6.8 | 19.4 ± 5.0 | 14.4 ± 3.0 | 63.6 ± 2.2 | **95.0 ± 0.9** | / | 91 ± 1 |
| humanoidmaze-large-navigate-v0 | 48.0 ± 4.7 | **79.4 ± 5.6** | 2.1 ± 1.1 | 8.4 ± 3.6 | 6.8 ± 2.0 | 10.0 ± 2.7 | 46.8 ± 6.8 | 64.3 ± 6.0 | / | / |
| humanoidmaze-large-stitch-v0 | 28.6 ± 2.9 | 65.1 ± 10.7 | 0.7 ± 0.5 | 2.0 ± 0.7 | 3.9 ± 2.5 | 3.1 ± 1.5 | 11.6 ± 5.3 | **75.6 ± 12.1** | / | **72 ± 3** |
| humanoidmaze-giant-navigate-v0 | 16.0 ± 8.6 | **85.3 ± 6.1** | 0.7 ± 0.3 | 0.5 ± 2.0 | 1.1 ± 0.5 | 1.0 ± 1.0 | 40.4 ± 2.7 | 79.0 ± 5.3 | 14.0 ± 5.3 | / |
| humanoidmaze-giant-stitch-v0 | 4.4 ± 1.3 | **78.1 ± 5.1** | 0.2 ± 0.3 | 0.2 ± 0.5 | 0.5 ± 0.4 | 4.1 ± 2.4 | 0.0 ± 0.1 | **79.8 ± 5.6** | 8.3 ± 4.5 | 67 ± 4 |
| visual-antmaze-large-navigate-v0 | 72.0 ± 3.1 | **83.8 ± 2.6** | 2.4 ± 0.6 | 3.6 ± 0.6 | 0.9 ± 2.4 | 0.3 ± 0.7 | 72.2 ± 4.6 | 75.8 ± 5.2 | 85.2 ± 6.6 | / |
| visual-antmaze-giant-navigate-v0 | 5.2 ± 5.2 | 7.6 ± 7.7 | 0.5 ± 0.4 | 0.2 ± 0.3 | 0.1 ± 0.3 | 3.3 ± 3.2 | 4.2 ± 1.2 | 8.4 ± 4.0 | 67.2 ± 3.0 | / |
| visual-antmaze-large-stitch-v0 | 28.7 ± 5.5 | 66.4 ± 16.0 | 0.1 ± 0.3 | 0.0 ± 0.0 | 0.4 ± 0.5 | 0.4 ± 0.6 | 6.6 ± 7.6 | 60.4 ± 36.4 | 77.2 ± 6.1 | / |
| visual-antmaze-giant-stitch-v0 | 0.2 ± 0.4 | 32.2 ± 17.3 | 0.0 ± 0.0 | 0.0 ± 0.0 | 0.0 ± 0.1 | 6.0 ± 3.1 | 0.0 ± 0.0 | 31.0 ± 24.9 | 51.9 ± 6.4 | / |
| visual-antmaze-medium-explore-v0 | 0.2 ± 0.6 | **63.0 ± 28.1** | 0.0 ± 0.0 | 0.0 ± 0.0 | 0.7 ± 1.0 | 0.8 ± 1.5 | 2.4 ± 3.4 | 58.0 ± 26.7 | 65.9 ± 6.8 | / |
| visual-antmaze-large-explore-v0 | 0.0 ± 0.0 | 0.8 ± 1.6 | 0.0 ± 0.0 | 0.0 ± 0.0 | 0.0 ± 0.0 | 0.0 ± 0.0 | 0.0 ± 0.0 | **15.2 ± 13.5** | 15.1 ± 6.8 | / |
| visual-humanoidmaze-medium-navigate-v0 | **0.8 ± 0.9** | 0.4 ± 0.5 | 0.0 ± 0.0 | 0.0 ± 0.0 | 0.0 ± 0.0 | 0.0 ± 0.0 | 0.1 ± 0.2 | 0.2 ± 0.2 | 0.3 ± 0.2 | / |
| visual-humanoidmaze-medium-stitch-v0 | 0.4 ± 0.5 | **0.7 ± 0.6** | 0.0 ± 0.0 | 0.0 ± 0.0 | 0.0 ± 0.0 | 0.0 ± 0.0 | 0.2 ± 0.3 | 0.2 ± 0.4 | 0.2 ± 0.2 | / |

**Findings.** Despite requiring no additional training, TTGS consistently improves the performance of competent base learners and, on several long-horizon tasks ~~especially in `pointmaze` and~~ especially in `humanoidmaze` and the giant variants of `pointmaze`, outperforms more complex planning baselines that rely on extra training. These gains support our central claim: simple metric-guided test-time planning can unlock long-horizon competence already latent in value-based GCRL agents.

**Reproduction details and caveats.** GAS paper reports only `antmaze` results, so we ran the authors' official implementation for all domains, using their `antmaze` hyperparameters, over 8 random seeds. CompDiffuser results are taken directly from Luo et al. (2025).

**Comparison with GAS using fixed hyperparameters.** To ensure a fair comparison with GAS, which uses hyperparameters tuned on `antmaze`, we evaluate HIQL+TTGS using a single fixed set of hyperparameters ($\tau = 24, T = 48$) derived from `antmaze` across all domains. Results are reported in Table 4. While GAS achieves higher scores on `antmaze`, TTGS outperforms GAS on `humanoidmaze` and `pointmaze`, highlighting the robustness of TTGS across diverse domains despite using fixed hyperparameters.

Table 4: Success rates on OGBench locomotion tasks for HIQL, HIQL+TTGS using a single fixed set of TTGS hyperparameters derived from `antmaze` ($\tau = 24, T = 48$), and GAS. While GAS achieves higher scores on `antmaze` (the domain it is tuned for), TTGS with fixed hyperparameters drastically outperforms GAS on `humanoidmaze` and `pointmaze`, highlighting TTGS as a more robust cross-domain planner.

| Dataset | HIQL | HIQL+TTGS (Fixed) | GAS |
|---|---|---|---|
| pointmaze-giant-navigate | $47 \pm 10$ | $\mathbf{68 \pm 13}$ | $5 \pm 11$ |
| pointmaze-giant-stitch | $0 \pm 0$ | $\mathbf{73 \pm 14}$ | $0 \pm 0$ |
| antmaze-giant-navigate | $67 \pm 4$ | $68 \pm 3$ | $\mathbf{76 \pm 6}$ |
| antmaze-giant-stitch | $2 \pm 1$ | $48 \pm 19$ | $\mathbf{86 \pm 4}$ |
| antmaze-large-explore | $5 \pm 7$ | $60 \pm 26$ | $\mathbf{91 \pm 9}$ |
| humanoidmaze-giant-navigate | $14 \pm 6$ | $\mathbf{78 \pm 9}$ | $14 \pm 5$ |
| humanoidmaze-giant-stitch | $4 \pm 2$ | $\mathbf{69 \pm 18}$ | $8 \pm 4$ |

## B  VALUE FUNCTION TO DISTANCE MAPPINGS

**Value-to-distance transformation for HIQL, SAW, and GCIQL.** ~~Both HIQL~~ HIQL, SAW, and GCIQL use a per-step penalty, that is reward $-1$ until the goal and $0$ at the goal. As discussed in Section 3.1, we map goal-conditioned values to predicted step counts via

$$d(s, g) = \log_\gamma\big(1 + (1 - \gamma)\, V^*(s, g)\big),$$

where $\gamma \in (0, 1)$ is the discount factor. To avoid infinite outputs, we clip value predictions to the open interval $\left(-\frac{1}{1-\gamma}, 0\right)$ with a small margin $\varepsilon = 10^{-3}$:

$$V_{\text{clip}}(s, g) = \min\big(-\varepsilon,\ \max\big(V(s, g),\ -\tfrac{1}{1-\gamma} + \varepsilon\big)\big).$$

We then apply the transform to $V_{\text{clip}}$. For HIQL and SAW we average the two value heads, $V(s, g) = \frac{1}{2}\big(V_1(s, g) + V_2(s, g)\big)$. For GCIQL we use its single value head $V(s, g)$.

**QRL (quasi-metric) distance.** Quasi-metric RL learns a nonnegative function $d_{\text{qrl}}(s, g)$ that approximates the number of steps from $s$ to $g$. This head already represents a distance, so we use it directly without additional transformation.

**Minimum step constraint.** In all cases we lower bound predicted distances by 1 step, since no transition can be completed with fewer than one action.

## C ~~ALTERNATIVE TRAINING DATA FILTERING~~ MANIPULATION TASKS AND VALUE FUNCTION GEOMETRY

We ~~tested a vertex sampling scheme inspired by Baek et al. (2025). It did not beat uniform random sampling in preliminary runs and added complexity, so we report it only for completeness.~~ evaluate TTGS on OGBench manipulation tasks in Table 5, pairing it with the strongest base agent for these domains, GCIQL. We observe that improvements are consistent but modest compared to locomotion tasks.

Table 5: **Success rates on OGBench manipulation tasks.** Improvements are modest because manipulation datasets lack intermediate states connecting start and evaluation goals, so TTGS often correctly defaults to the base policy instead of hallucinating unsupported plans.

| Dataset | GCIQL Success | GCIQL+TTGS Success |
|---|---|---|
| scene-play | $50 \pm 7$ | $52 \pm 4$ |
| cube-triple-play | $4 \pm 2$ | $4 \pm 2$ |
| puzzle-4x5-play | $12 \pm 3$ | $13 \pm 3$ |
| puzzle-4x6-play | $10 \pm 0$ | $10 \pm 0$ |

~~Given trajectories $\tau = (s_0, \ldots, s_T)$ with terminal flags and a lookahead $H \in \mathbb{N}$, we score states by temporal efficiency using the agent's distance predictor $\hat{d}$ and then cluster the kept states. We also investigated clustering uniformly selected states.~~ To understand this behavior, we analyze the geometry of the learned value functions. In Figure 5, we visualize the distribution of the sampled dataset states on a 2D plane defined by the predicted distance from the start state ($x$-axis) and the predicted distance to the goal ($y$-axis).

~~**Temporal efficiency.** For time step $t$ and horizon $h$ with $t + h \leq T$, define~~

$$\mathrm{H_{TE}}(s_t) = \frac{\hat{d}(s_t, s_{t+h})}{h}.$$

~~When $\hat{d}$ is calibrated in steps, values near one indicate accurate local progress. Keep states with~~

$$1 - \varepsilon \leq \mathrm{H_{TE}}(s_t) \leq 1 + \varepsilon \quad (\varepsilon = 0.005 \text{ in our runs}).$$

In locomotion tasks (`humanoidmaze`, `antmaze`), states typically form a curve connecting the start and goal (low $x + y$), providing a dense set of intermediate subgoals for the graph search. In contrast, for manipulation tasks (`cube-triple`, `scene`), the dataset states often cluster far from both the start and the evaluation goal, with very few states bridging the gap. This indicates that the evaluation goals lie outside the main data manifold. Crucially, TTGS correctly identifies that no reliable path exists through the graph—due to high edge costs across the manifold gap—and defaults to the base policy behavior. This confirms that TTGS acts as a safe planning wrapper that does not hallucinate plans when data support is missing.

## D ALTERNATIVE TRAINING DATA FILTERING

We tested a vertex sampling scheme based on clustering to improve state coverage, inspired by Baek et al. (2025). While Baek et al. (2025) filter states by temporal efficiency before clustering, we found that adapting this metric to standard value functions was ineffective in preliminary runs. Consequently, we compare standard TTGS (which uses uniform random sampling for graph vertices) against a variant that selects vertices by clustering a large pool of randomly sampled states.

**Clustering.** ~~Let~~ We sample a large subset $\mathcal{H}$ ~~be the kept states. Build centers~~ of 80,000 states uniformly from $\mathcal{D}$. We select graph vertices $\mathcal{V} \subset \mathcal{H}$ ~~by~~ using a single-pass greedy clustering rule with radius ~~$r$ (for example $r = h/2$)~~ $r = \tau/2$:

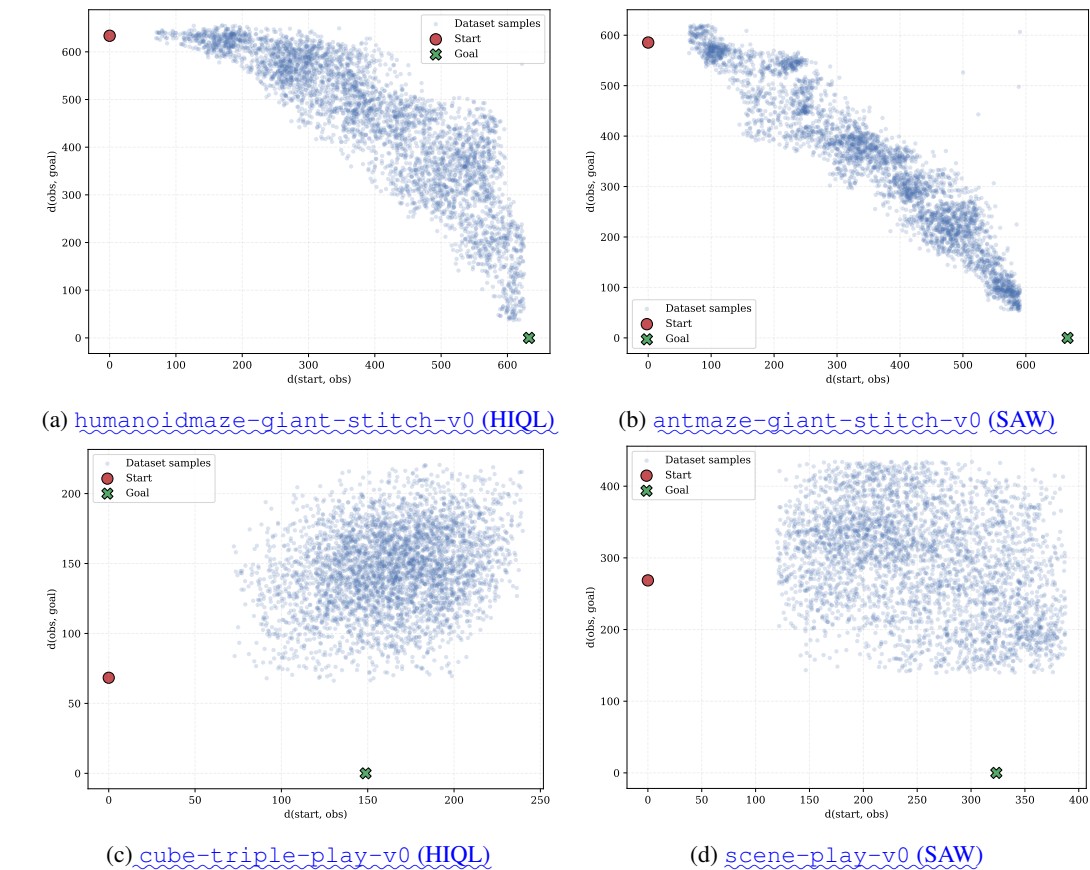

(a) `humanoidmaze-giant-stitch-v0` (HIQL)      (b) `antmaze-giant-stitch-v0` (SAW)

(c) `cube-triple-play-v0` (HIQL)      (d) `scene-play-v0` (SAW)

Figure 5: **Value Function Geometry.** In locomotion tasks (a, b), dataset states form a connected band between start and goal (low $x + y$), supporting planning. In manipulation tasks (c, d), evaluation goals often lie far from the data distribution, creating a gap where no intermediate states exist. TTGS correctly identifies these high-cost edges and defaults to the base policy rather than hallucinating paths.

1. Initialize $\mathcal{V} \leftarrow \{s^{(0)}\}$, the first element of $\mathcal{H}$.

2. For each $x \in \mathcal{H}$, compute $m = \min_{v \in \mathcal{V}} \hat{d}(x, v)$.

3. If $m > r$, add $x$ to $\mathcal{V}$. Otherwise assign $x$ to its nearest center and periodically update that center to the member that minimizes total within-cluster distance.

Table 6 compares the performance and runtime of TTGS with random sampling versus clustering. We observe no statistically significant performance gain from clustering, while the computational cost of graph construction increases by an order of magnitude (from ∼40 seconds to >10 minutes). Thus, we adopt random sampling as the default for its efficiency.

Table 6: **Effect of state clustering on TTGS.** We compare TTGS with and without state clustering. Clustering uses a random subset of 80,000 states, clustering distance threshold $\frac{1}{2}\tau$, and a maximum of 4,000 cluster centers. From the empirical results, clustering provides only marginal gains while its computational overhead is dominant in the overall cost.

| Environment | HIQL | HIQL+TTGS w/o Clustering | HIQL+TTGS w/ Clustering | Clustering Time (s) | Cluster Centers |
|---|---|---|---|---|---|
| pointmaze-giant-navigate-v0 | $43 \pm 12$ | $73 \pm 12$ | $74 \pm 9$ | 814.76 | 613.0 |
| pointmaze-giant-stitch-v0 | $0 \pm 0$ | $80 \pm 6$ | $79 \pm 10$ | 876.43 | 3904.88 |
| antmaze-giant-navigate-v0 | $64 \pm 4$ | $67 \pm 3$ | $69 \pm 3$ | 706.77 | 4000.0 |
| antmaze-giant-stitch-v0 | $2 \pm 1$ | $70 \pm 14$ | $69 \pm 11$ | 1932.88 | 4000.0 |
| antmaze-large-explore-v0 | $2 \pm 4$ | $26 \pm 34$ | $25 \pm 33$ | 771.11 | 4000.0 |
| humanoidmaze-giant-navigate-v0 | $15 \pm 7$ | $85 \pm 6$ | $83 \pm 9$ | 691.54 | 4000.0 |
| humanoidmaze-giant-stitch-v0 | $4 \pm 2$ | $78 \pm 8$ | $78 \pm 12$ | 705.48 | 4000.0 |

# E    SOFT VS. HARD EDGE PENALTIES

We investigate whether the soft penalty scheme is necessary compared to a simpler hard thresholding scheme. We compare our method (Soft Penalty) against a variant where edges with predicted distance greater than $\tau$ are removed entirely (Hard Threshold).

Table 7 presents the results. On simple navigation tasks, the Hard Threshold often performs comparably to the Soft Penalty. However, on tasks requiring trajectory stitching, the Hard Threshold leads to drastic performance drops. For example, success drops from 78.6% to 2.2% on antmaze-giant-stitch-v0 and from 78.1% to 14.0% on humanoidmaze-giant-stitch-v0.

This performance gap is explained by the *Disconnect Ratio*, defined as the fraction of episodes where the goal becomes unreachable from the start state in the constructed graph. Standard value functions are noisy and may underestimate the cost of specific transitions required to bridge disparate trajectories in the dataset. A hard threshold disconnects these essential links, rendering planning impossible. The soft penalty allows these edges to persist with high cost, preserving connectivity while still discouraging their use unless no safer path exists.

Table 7: **Comparison of Soft vs. Hard Penalties.** The Hard Threshold variant frequently disconnects the graph on stitching tasks (high Disconnect Ratio), leading to poor success rates. The Soft Penalty maintains connectivity by assigning high costs to optimistic edges rather than removing them. Results are for HIQL+TTGS.

| Dataset | HIQL+TTGS (Soft) | HIQL+TTGS (Hard) | Disconnect Ratio |
|---|---|---|---|
| pointmaze-giant-navigate-v0 | $71 \pm 12$ | $65 \pm 10$ | 0.08 |
| pointmaze-giant-stitch-v0 | $81 \pm 9$ | $66 \pm 9$ | 0.25 |
| antmaze-giant-navigate-v0 | $66 \pm 4$ | $67 \pm 4$ | 0.11 |
| antmaze-giant-stitch-v0 | $79 \pm 13$ | $2 \pm 2$ | 0.99 |
| antmaze-large-explore-v0 | $27 \pm 34$ | $53 \pm 32$ | 0.00 |
| humanoidmaze-giant-navigate-v0 | $85 \pm 6$ | $57 \pm 9$ | 0.47 |
| humanoidmaze-giant-stitch-v0 | $78 \pm 5$ | $14 \pm 14$ | 0.84 |
| visual-antmaze-large-navigate-v0 | $84 \pm 3$ | $64 \pm 9$ | 0.38 |
| visual-antmaze-large-stitch-v0 | $66 \pm 16$ | $31 \pm 14$ | 0.54 |

## F    RUNTIME ANALYSIS

We provide a detailed breakdown of the computational overhead of TTGS in Table 8. All runtimes were measured on a single Nvidia L40s GPU with 8 virtual CPU cores. Graph construction is performed once per dataset, and the shortest path search is performed once per episode (or sparingly if replanning). Both operations are matrix-heavy and highly parallelizable; we implement them on the GPU. The per-episode planning overhead is generally around 1 second or less. The per-step action selection overhead is negligible ($\approx 1.5$ ms).

Table 8: **TTGS Runtime Overhead.**  Per-environment cost of graph construction and online planning components in TTGS. Measured on an Nvidia L40s GPU with 8 vCPUs using HIQL as base agent. Graph build is one-time compute for all the tasks, while shortest path is one-time compute per episode.

| Environment | Graph Construction (s) | Shortest Path (s) | Pick Subgoal (ms) | Step Overhead (ms) |
|---|---|---|---|---|
| pointmaze-giant-navigate-v0 | 36.42 | 1.25 | 0.36 | 1.68 |
| pointmaze-giant-stitch-v0 | 34.98 | 0.79 | 0.37 | 1.61 |
| antmaze-giant-navigate-v0 | 36.55 | 0.46 | 0.37 | 1.42 |
| antmaze-giant-stitch-v0 | 36.26 | 0.84 | 0.39 | 1.66 |
| antmaze-large-explore-v0 | 36.40 | 0.46 | 0.38 | 1.33 |
| humanoidmaze-giant-navigate-v0 | 34.99 | 0.59 | 0.38 | 1.39 |
| humanoidmaze-giant-stitch-v0 | 36.35 | 1.0 | 0.39 | 1.38 |
| visual-antmaze-large-navigate-v0 | 100.02 | 0.53 | 0.47 | 2.63 |
| visual-antmaze-large-stitch-v0 | 100.89 | 0.53 | 0.48 | 2.49 |

## G    REPLANNING ANALYSIS

We evaluate a variant of TTGS that triggers a full recalculation of the shortest path if the agent deviates from the current subgoal by more than $2T$. To prevent excessive path oscillations, we limit replanning to occur at most once every 50 steps. Table 9 compares this against the standard one-time planning approach. Results show that while replanning can help in datasets with highly noisy trajectories (e.g., antmaze-large-explore), the adaptive subgoal selection on the fixed guide path is sufficient and often superior for most tasks.

Table 9: **Effect of replanning in TTGS.** Comparison between the default TTGS (one-time planning) with a variant that replans when the agent strays $> 2T$ from the current subgoal (at most once every 50 steps). *Replan Episode Ratio* indicates the fraction of episodes where replanning occurred. In most environments, the one-time adaptive guide is sufficient, and frequent replanning can degrade performance by causing oscillations.

| Environment | HIQL | HIQL+TTGS w/o Replan | HIQL+TTGS w/ Replan | Replan Episode Ratio |
|---|---|---|---|---|
| pointmaze-giant-navigate-v0 | 50 ± 13 | **73 ± 12** | **72 ± 6** | 0.01 |
| pointmaze-giant-stitch-v0 | 0 ± 0 | **80 ± 6** | **80 ± 6** | 0.00 |
| antmaze-giant-navigate-v0 | 66 ± 3 | **67 ± 3** | **71 ± 2** | 0.52 |
| antmaze-giant-stitch-v0 | 1 ± 1 | **70 ± 14** | **67 ± 13** | 0.38 |
| antmaze-large-explore-v0 | 2 ± 3 | 26 ± 34 | **50 ± 32** | 0.07 |
| humanoidmaze-giant-navigate-v0 | 14 ± 5 | **85 ± 6** | 78 ± 4 | 0.52 |
| humanoidmaze-giant-stitch-v0 | 4 ± 3 | **78 ± 8** | 33 ± 10 | 0.98 |
| visual-antmaze-large-navigate-v0 | 46 ± 14 | **82 ± 7** | 70 ± 10 | 0.86 |
| visual-antmaze-large-stitch-v0 | 18 ± 8 | **78 ± 4** | 53 ± 24 | 0.76 |

## H   HYPERPARAMETERS

TTGS does not require training and has only three hyperparameters, so tuning is quick and simple. We did not run extensive sweeps; Table 10 lists the values used in our experiments.

$M$ is the number of dataset states sampled as graph vertices. Larger $M$ increases coverage and cost. In our preliminary experiments we found $M = 4000$ to be sufficient while performant, with $M$ as low as 100 helpful for improving base agent's performance.

$\tau$ is the edge-length threshold used to compute graph weights: distances greater than $\tau$ are penalized. A larger $\tau$ permits longer hops and is suitable only when long-range distance estimates are accurate and the pretrained policy can reliably traverse them.

$T$ is the maximum allowed distance between the current state and the selected subgoal at execution time. Choose $T$ near the range where the frozen policy is most reliable.

## I   USE OF GENERATIVE AI

LLMs were used to revise and polish writing on a single-paragraph scale.

Table 10: Hyperparameter settings used in our experiments.

| Dataset~~Type~~ | ~~Base Agent~~ $M$ | ~~$\tau$~~ HIQL $(\tau, T)$ | ~~$T$~~ QRL $(\tau, T)$ | GCIQL $(\tau, T)$ | SAW $(\tau, T)$ |
|---|---|---|---|---|---|
| pointmaze-medium-navigate-v0 | ~~HIQL~~ 4000 | (24, 48) | (24, 48) | ~~QRL 4000~~ (24, 48) | (24, 48) |
| pointmaze-medium-stitch-v0 | ~~GCIQL~~ 4000 | ~~12~~ (24, 48) | (24, 48) | ~~HIQL~~ (24, 48) | ~~4000 12~~ (24, 48) |
| pointmaze-large-navigate-v0 | ~~QRL~~ 4000 | ~~12~~ (24, 48) | (24, 48) | ~~GCIQL~~ (24, 48) | ~~4000 6 12~~ (24, 48) |
| pointmaze-large-stitch-v0 | ~~HIQL~~ 4000 | (24, 48) | (24, 48) | ~~QRL~~ (24, 48) | ~~4000 12~~ (24, 48) |
| pointmaze-giant-navigate-v0 | ~~GCIQL~~ 4000 | ~~12~~ (24, 48) | (24, 48) | ~~HIQL 4000~~ (12, 24) | (24, 48) |
| pointmaze-giant-stitch-v0 | ~~QRL~~ 4000 | (12, 24) | (12, 24) | ~~GCIQL 4000~~ (6, 12) | (24, 48) |
| antmaze-medium-navigate-v0 | ~~HIQL~~ 4000 | (24, 48) | (24, 48) | ~~QRL 4000~~ (24, 48) | (24, 48) |
| antmaze-medium-stitch-v0 | ~~GCIQL~~ 4000 | (24, 48) | (24, 48) | ~~HIQL~~ (24, 48) | ~~4000 36 72~~ (24, 48) |
| antmaze-large-navigate-v0 | ~~QRL~~ 4000 | (24, 48) | (24, 48) | ~~GCIQL 4000~~ (24, 48) | (24, 48) |
| antmaze-large-stitch-v0 | ~~HIQL~~ 4000 | (24, 48) | (24, 48) | ~~QRL 4000~~ (24, 48) | (24, 48) |
| antmaze-giant-navigate-v0 | ~~GCIQL~~ 4000 | (24, 48) | (12, 24) | (12, 24) | (24, 48) |
| antmaze-giant-stitch-v0 | ~~HIQL~~ 4000 | (12, 24) | (12, 24) | ~~QRL 4000~~ (12, 24) | (24, 48) |
| antmaze-large-explore-v0 | ~~GCIQL~~ 4000 | ~~12~~ (24, 48) | (24, 48) | ~~HIQL~~ (24, 48) | (24, 48) |
| humanoidmaze-medium-navigate-v0 | 4000 | ~~12~~ (24, 48) | (24, 48) | ~~QRL~~ (24, 48) | (24, 48) |
| humanoidmaze-medium-stitch-v0 | 4000 | ~~12~~ (24, 48) | (24, 48) | ~~GCIQL~~ (24, 48) | (24, 48) |
| humanoidmaze-large-navigate-v0 | 4000 | ~~12~~ (24, 48) | (24, 48) | ~~HIQL~~ (24, 48) | (24, 48) |
| humanoidmaze-large-stitch-v0 | 4000 | ~~12~~ (24, 48) | (24, 48) | ~~QRL~~ (24, 48) | (24, 48) |
| humanoidmaze-giant-navigate-v0 | 4000 | ~~12~~ (36, 72) | (24, 48) | ~~GCIQL~~ (24, 48) | (24, 48) |
| humanoidmaze-giant-stitch-v0 | 4000 | (24, 48) | (24, 48) | (12, 24) | (24, 48) |
| visual-antmaze-large-navigate-v0 | ~~HIQL~~ 4000 | (12, 24) | (12, 24) | ~~QRL 4000~~ (12, 24) | (24, 48) |
| visual-antmaze-giant-navigate-v0 | ~~GCIQL~~ 4000 | (12, 24) | (12, 24) | ~~HIQL 4000~~ (12, 24) | (24, 48) |
| visual-antmaze-large-stitch-v0 | ~~QRL~~ 4000 | (12, 24) | (12, 24) | ~~GCIQL 4000~~ (12, 24) | (24, 48) |
| visual-antmaze-giant-stitch-v0 | ~~HIQL~~ 4000 | (12, 24) | (12, 24) | ~~QRL 4000~~ (12, 24) | (24, 48) |
| visual-antmaze-medium-explore-v0 | ~~GCIQL~~ 4000 | (12, 24) | (12, 24) | ~~HIQL 4000~~ (12, 24) | (24, 48) |
| visual-antmaze-large-explore-v0 | ~~QRL~~ 4000 | (12, 24) | (12, 24) | ~~GCIQL 4000~~ (12, 24) | (24, 48) |
| visual-humanoidmaze-medium-navigate-v0 | ~~HIQL~~ 4000 | (12, 24) | (12, 24) | ~~QRL 4000~~ (12, 24) | (24, 48) |
| visual-humanoidmaze-medium-stitch-v0 | ~~GCIQL~~ 4000 | (12, 24) | (12, 24) | (12, 24) | (24, 48) |

