# OpenReview forum: "Test-Time Graph Search for Goal-Conditioned Reinforcement Learning"
_ICLR.cc/2026/Conference — Submitted to ICLR 2026_

### Official Review · Reviewer_pdK4 · 2025-10-26

**Soundness:** 2
**Presentation:** 3
**Contribution:** 1
**Rating:** 2
**Confidence:** 3

**Summary:**

This work introduces a test-time routine for goal-conditioned reinforcement learning. By relying on (often available) pre-training dataset and an estimator of temporal distances (e.g. the goal-conditioned critic), the algorithm constructs a graph whose vertices are states uniformly sampled from the dataset. Each edge is weighted according to the (potentially learned) distance between its two endpoints; long edges are penalized in order to compensate for estimation errors. Dijkstra's algorithm can then find the shortest path from the current state to the goal over the graph: the next vertex on the path can be commanded to the policy as a subgoal. The authors provide evidence for the decay in current algorithms' performance on distant goals (Figure 2) as a motivation of this work. The method is then evaluated in giant versions of diverse OGBench mazes. Finally, the authors ablate the criterion for subgoal selection, as well as relevant hyperparameters.

**Strengths:**

- This work is clearly presented and well-written.
- The motivation is clear; Figure 2 is instrumental to highlighting it.
- The experiments are well designed and informative with respect to the method's effectiveness.

**Weaknesses:**

- The main weakness of this work lies, in my opinion, in its novelty. Methodologically, the proposed technique matches Search on The Replay Buffer (Eysenbach 2019), among many other works citing it and predating it. To the best of my knowledge, the main novelty is in the soft penalization of edge weights, which however is exponential and therefore rather hard. It is not clear whether this should be a fundamentally better option than the standard hard penalization scheme. The claim on lines 52-53 ("these approaches require specialized training or additional data") does not apply to SORB to the best of my knowledge.
- The evaluation in this work is limited to mazes, in which semi-parametric techniques are known to work well, as reported in SORB. The real challenge of these approaches lies, in my opinion, in less structured environment (e.g. manipulation). While well-planned, the experimental section does not report significantly new findings.

**Questions:**

## Minor issues and questions
- Why do self-loops need to be penalized (line 245)? Is there a reason for Dijkstra to select self-loops?
- Why is GCIQL outperforming HIQL in Figure 2c?
- Line 312: the comparison with HIQL is not a clean ablation of learned vs non-learned high-level planner. HIQL uses a single n-steps ahead value estimate to guide subgoal selection, while the proposed method performs full planning on the graph, which is inherently more capable.
- Line 441: could hierarchical clustering help in reducing the number of pairwise distance to be computed? This might be an interesting extension in case compute is seen as a limitation.
- The performance of GAS on pointmaze in Table 3 stands out in the column. Is there a reason why the method performs poorly?

## Conclusion
While this submission is well written and well designed, I believe that it does not make a significant (methodological or empirical) contribution on top of existing graph-search-based planning methods such as SORB. In case this stems from a misunderstanding, I am happy to discuss this point at length in the rebuttal.

---

> ### Author Response · Authors · 2025-11-20
> **Official Reply to Reviewer pdK4 [1/2]**
>
> We thank the reviewer for their valuable feedback and for acknowledging the clarity of our presentation, the strength of our motivation, and the quality of our experimental design. We address your concerns regarding novelty, edge penalization, and generality below.
>
> > Methodologically, the proposed technique matches Search on The Replay Buffer (SORB, Eysenbach 2019)... The claim ... ("these approaches require specialized training or additional data") does not apply to SORB...
>
> We respectfully clarify that while TTGS and SORB share the high-level intuition of graph search, they operate under fundamentally different constraints and inputs. **SORB is a full RL algorithm** that learns a distance function from scratch, whereas **TTGS is a post-hoc inference wrapper** that repurposes existing value functions.
>
> We address a common scenario: a practitioner has a pretrained GCRL agent (e.g., GCIQL, HIQL) that excels at short horizons but fails on long ones. TTGS offers a "plug-in" solution to rescue this policy without modifications to the base agent or additional training. This distinguishes TTGS from SORB in three key aspects:
>
> 1.  **Different Problem Setting:** SORB is an *online* algorithm designed to improve exploration and data collection during training. TTGS is a purely *offline, test-time* method. It operates on a fixed dataset and a frozen policy, a constrained setting highly relevant to real-world deployment where retraining is expensive or unsafe.
> 2. **Training Requirements:** SORB relies on specific training choices to work. Specifically, SORB requires:
>     *   **Distributional Q-Learning:** It discretizes the value function into distance bins and uses a custom "left-shift" Bellman update tailored to the `-1` per-step reward structure.
>     *   **Ensembles:** For image-based tasks, SORB’s authors state that training an ensemble of Q-networks was "crucial" for success (Section 3.2), confirmed by their ablations (Section 5.4).
>
>     In contrast, TTGS is agnostic to the training method. It consumes scalar value estimates from off-the-shelf agents like HIQL, GCIQL, or SAW (as shown in reply to Reviewer XeSt).
>
> 3.  **Generality:** TTGS demonstrates a novel scientific finding: standard value functions (trained via expectile regression or contrastive learning, not for planning) already contain sufficient geometric structure to support graph search if processed correctly (via our soft penalty). SORB assumes specialized training is necessary; we show it is not.
>
> The contribution of TTGS is demonstrating that standard value functions, trained only for policy learning, not for planning, contain sufficient geometric information to enable graph search without the specialized machinery of SORB.
>
>
> > ...novelty is in the soft penalization of edge weights... It is not clear whether this should be a fundamentally better option than the standard hard penalization scheme.
>
> The soft penalty is essential when using *standard* value functions as distance metrics. Unlike SORB, which explicitly learns a distance function, standard value functions are often noisy. A hard cutoff may disconnect the graph when the value function underestimates a single reachable transition, rendering planning impossible.
>
> To demonstrate this, we compared our soft penalty against a hard threshold (removing edges $>\tau$).
>
> | Environment | TTGS (Soft Threshold) | TTGS (Hard Threshold) | Disconnect Ratio
> | --- | --- | --- | --- |
> | antmaze-giant-navigate | 67 ± 3 | 67 ± 4 | 0.11 |
> | antmaze-giant-stitch | 70 ± 14 | 2 ± 2 | 0.99 |
> | antmaze-large-explore | 26 ± 34 | 53 ± 32 | 0.00 |
> | humanoidmaze-giant-navigate | 85 ± 6 | 57 ± 9 | 0.47 |
> | humanoidmaze-giant-stitch | 78 ± 8 | 14 ± 14 | 0.84 |
> | pointmaze-giant-navigate | 73 ± 12 | 65 ± 10 | 0.08 |
> | pointmaze-giant-stitch | 80 ± 6 | 66 ± 9 | 0.25 |
> | visual-antmaze-large-navigate | 82 ± 7 | 64 ± 9 | 0.38 |
> | visual-antmaze-large-stitch | 78 ± 4 | 31 ± 14 | 0.54 |
>
>
> As shown in the "Disconnect Ratio" column, the Hard Penalty frequently severs the path to the goal (e.g., in 99% of `antmaze-giant-stitch` episodes). The Soft Penalty maintains connectivity by allowing "optimistic" edges but assigning them high cost, discouraging their use unless no safer path exists.

---

> ### Author Response · Authors · 2025-11-20
> **Official Reply to Reviewer pdK4 [2/2]**
>
> > The evaluation in this work is limited to mazes... real challenge ... lies in less structured environment (e.g. manipulation).
>
> As detailed in our response to Reviewer z4AS, we have expanded our evaluation to include OGBench manipulation tasks, pairing TTGS with GCIQL.
>
> | Environment | GCIQL Success | GCIQL+TTGS Success |
> | :--- | :--- | :--- |
> | scene-play | 50 ± 7 | 52 ± 4 |
> | cube-triple-play | 4 ± 2 | 4 ± 2 |
> | puzzle-4x5-play | 12 ± 3 | 13 ± 3 |
> | puzzle-4x6-play | 10 ± 0 | 10 ± 0 |
>
> Improvements are consistent but modest. We analyzed the geometry of the value function and found a key data property: unlike locomotion, where states bridge the start and goal, manipulation datasets often lack intermediate states connecting to the evaluation goals. In these cases, TTGS correctly identifies that no low-cost path exists through the graph and defaults to the base policy behavior. This confirms that TTGS is robust: it does not "hallucinate" plans when the data support is missing. We will add this table and analysis to the paper.
>
> > Why do self-loops need to be penalized (line 245)? Is there a reason for Dijkstra to select self-loops?
>
> Since we lower-bound predicted distances by 1 step (as a transition requires at least one action), a self-loop would always have positive cost and would never be selected by an optimal shortest-path algorithm. Explicitly removing them is a graph optimization to reduce edge count, not a change in logic.
>
> > Why is GCIQL outperforming HIQL in Figure 2c?
>
> Figure 2c illustrates the general trend of performance decay over long horizons for all agents, it is not intended to compare them. We hypothesise that GCIQL is a strong "flat" policy, which sometimes dominates on short horizons where hierarchy introduces unnecessary overhead.
>
> > the comparison with HIQL is not a clean ablation of learned vs non-learned high-level planner... the proposed method ... is inherently more capable.
>
> We agree. Our central argument is precisely that learned high-level planners (like HIQL's) struggle with long-horizon consistency compared to explicit graph search. Our contribution is showing *how* to inject this "inherently more capable" planning into existing agents without retraining them.
>
> > could hierarchical clustering help in reducing the number of pairwise distance to be computed?
>
> We tested a clustering approach (Appendix C) but found it offered no performance gain over uniform sampling for a fixed node budget, while increasing graph construction time from $\approx 36$s to $>10$ minutes. We performed an additional experiment, detailed in reply to Reviewer z4AS which also indicates that clustering does not provide a statistically significant improvement for tested environments. We will update Appendix C with this new result.
>
> > The performance of GAS on pointmaze in Table 3 stands out... Is there a reason why the method performs poorly?
>
> We suspect GAS's learned representation struggles with the specific low-dimensional dynamics of `pointmaze`, or perhaps overfits to `antmaze` (where it excels). The OGBench authors also noted the surprisingly poor performance of several advanced methods on `pointmaze`. TTGS consistently offers a significant improvement in `pointmaze` tasks across all tested base agents and hyperparams.
>
> We hope these clarifications regarding the distinction from SORB and the necessity of our design choices address your concerns. Given the new data justifying our soft penalty and scope, would you be willing to reconsider your score? Do you require any additional information to make this decision?

---

> > ### Comment · Reviewer_pdK4 · 2025-11-22
> >
> > Thank you for taking the time to answer all of my questions, and the additional experiments outside of maze domains.
> >
> > I would respectfully disagree with your characterization of SORB.
> > - If you consider SORB as the full algorithm described in the original paper, then I understand that it's not an apple-to-apple comparison. However, if we just consider the "inference wrapper" component of SORB, I do not find any difference in terms of constraints and inputs. In GC-setting there is a clear bijection between values functions and temporal distances, independently from discounting and reward shifting.
> > - SORB does not *require* distributional update in its inference procedure, nor ensembles. To the best of my understanding, these are two additional practical contributions, which does not synergize with the inference procedure. In other words, the inference procedure of SORB is also agnostic to the training method, as shown by subsequent works that adapted it in different settings [1, 2]. These works also demonstrated that standard value functions may already support graph search.
> >
> > Regarding the soft penalization scheme, I understand the significance of the approach, and I think that the ablation provided is helpful. However, describing it as "essential" may be an overstatement, as a hard threshold seems to work nearly as well, or even better, on some domains. Furthermore, a hard threshold may also be computed on the fly to avoid disconnections altogether (the minimum threshold that ensures connection between the current state and goal can be computed efficiently through graph traversal).
> >
> > I appreciate the experiments in visual domains. The improvements are not significant, but that is understandable given the nature of these tasks. I also appreciate the transparent explanation of the algorithm behavior in these cases. Thank you for answering my other questions as well.
> >
> > For the moment being, I would keep my score, as my main concern still hold, and I do not follow the claims about the lack of generality of SORB.
> >
> > References:
> >
> > [1] Huang et al., Mapping State Space using Landmarks for Universal Goal Reaching, 2019
> >
> > [2] Kim et al., Imitating Graph Based Planning with Goal Conditioned Policies, 2023

---

> > > ### Author Response · Authors · 2025-11-25
> > > **Official reply to Reviewer pdK4 [1/2]**
> > >
> > > We thank the reviewer for the continued engagement and for the additional references. We appreciate the opportunity to clarify the specific gap TTGS fills and the empirical evidence supporting our design choices.
> > >
> > > > If you consider SORB as the full algorithm... it's not an apple-to-apple comparison. However, if we just consider the "inference wrapper" component of SORB, I do not find any difference... SORB does not require distributional update in its inference procedure...
> > >
> > > We agree that graph search over value functions is a shared high-level principle. However, the *scientific contribution* of TTGS is identifying the specific mechanism required to make this principle work for **standard, non-distributional, off-the-shelf agents**.
> > >
> > > The SORB paper (Eysenbach et al., 2019) explicitly identifies that standard Q-learning fails for graph search due to the "wormhole" problem (spurious shortcuts). Their solution was to change the **training** (Distributional RL). The papers you referenced support this difficulty: Huang et al. (2019) had to train specific *local* UVFAs because global values were unreliable, and Kim et al. (2023) introduced a self-imitation *training* loss to force alignment between policy and planner.
> > >
> > > In contrast, TTGS demonstrates that you *do not* need to change the training. You can take a standard HIQL or GCIQL checkpoint, trained without any knowledge of planning, and successfully apply graph search, but only if you use the **soft-penalty** mechanism in combination with **subgoal selection** procedure described in the paper.
> > >
> > > If one applies the "SORB inference wrapper" (standard graph search/hard threshold and next subgoal selection) to these standard agents, it fails on the hardest tasks. We show that with simple tweaks this idea can succeed in combination with pretrained GCRL agents.
> > >
> > > > describing [soft penalization] as "essential" may be an overstatement, as a hard threshold seems to work nearly as well, or even better, on some domains.
> > >
> > > We wish to highlight that the domains where the Hard Threshold works nearly as well are the easier `navigate` tasks. The only task where the mean with hard threshold is better is a noisy `explore` task. On the challenging `stitch` tasks, which are the primary motivation for long-horizon planning, the difference is catastrophic.
> > >
> > > Referring to the ablation table provided in our previous response:
> > > *   `antmaze-giant-stitch`: TTGS (Soft) **70%** vs. Hard Threshold **2%**.
> > > *   `humanoidmaze-giant-stitch`: TTGS (Soft) **78%** vs. Hard Threshold **14%**.
> > >
> > > A drop from ~70% to ~2% indicates that for standard value functions on complex long-horizon tasks, the soft penalty is indeed essential.
> > >
> > > Regarding the proposed dynamic hard threshold: while an interesting heuristic, it is untested in this context. It would require re-calculating connectivity for every start-goal pair at runtime (increasing inference cost), whereas our soft penalty is a static transformation computed once per dataset. The novelty lies in showing that this specific, static soft-weighting scheme is sufficient to unlock SOTA performance from noisy, standard value functions.
> > >
> > > > The improvements are not significant [in visual domains]
> > >
> > > While gains are smaller in `navigate` tasks (where the base agent is already capable), we highlight that in `visual-antmaze-large-stitch`, TTGS improves the base HIQL agent from 28.7% to 66.4% and in `visual-antmaze-medium-explore-v0` from  0.2% to 63%. We believe recovering the ability to stitch trajectories in visual domains without retraining is a significant result.
> > >
> > > > I do not follow the claims about the lack of generality of SORB
> > >
> > > Our claim is that TTGS is more *generally applicable* to existing model zoos.
> > > *   **SORB:** To use it, you must train your agent with Distributional RL + Ensembles.
> > > *   **TTGS:** You can download a pretrained HIQL, GCIQL, SAW, or QRL agent and immediately improve it.

---

> ### Author Response · Authors · 2025-11-25
> **Official reply to Reviewer pdK4 [2/2]**
>
> To demonstrate this generality and effectiveness, we have compiled a comparison of TTGS (applied to HIQL/GCIQL/QRL/SAW) against a wide range of baselines, including very recent concurrent works.
>
> | Environment | HIQL  | HIQL + TTGS | GCIQL | GCIQL+TTGS | QRL | QRL + TTGS | SAW | SAW+TTGS | CRL | OTA | TMD | Pi‑HIQL | CGCIVL | GC‑SSCP | TRA | Dual | GAS | CD | SCoTS | ProQ |
> | --- | --- | --- | --- | --- | --- | --- | --- | --- | --- | --- | --- | --- | --- | --- | --- | --- | --- | --- | --- | --- |
> | pointmaze-giant-navigate-v0 | 43±11 | 71±12 | 0±0 | **92±3** | 65±10 | 88±10 | 68±6 |**95±3** | 27±10 | 72±6 | / | 79±13 | 80±12 | 73±7 | / | / | 5±11 | / | / | **92 ± 3** |
> | pointmaze-giant-stitch-v0 | 0±0 | 81±9 | 0±0 | **98±1** | 55±12 | 93±8 | / | / | 0±0 | 52±7 | / | 22±10 | 81±17 | 0±0 | / | / | 0±0 | 68±3 | **100±0** |** 99 ± 1** |
> | antmaze-giant-navigate-v0 | 65±4 | 66±4 | 0±0 | 5±2 | 12±5 | 11±5 | 69±3 | 71±5 | 16±3 | **77±4** | / | 67±5 | 73±5 | 52±2 | 3±1 | 21±4 | **76±6** | / | / | / |
> | antmaze-giant-stitch-v0 | 1±1 | 79±13 | 0±0 | 33±7 | 2±3 | 52±24 | / | / | 0±0 | 37±6 | 2.7±3 | 48±11 | 36±7 | 0±0 | / | / | **86±4** | 65±3 | **87±2** | / |
> | humanoidmaze-giant-navigate-v0 | 16±9 | 85±6 | 1±0 | 1±2 | 1±1 | 1±1 | 40±3 | 79±5 | 3±2 | **92±1** | 9.2±1 | 68±5 | 29±9 | / | / | / | 14±5 | / | / | / |
> | humanoidmaze-giant-stitch-v0 | 4±1 | **78±5** | 0±0 | 0±1 | 1±0 | 4±2 | / | / | 0±0 | **79±3** | 6.3±1 | 19±5 | 34±6 | / | / | / | 8±6 | 67±4 | / | / |
> | visual-antmaze-large-navigate-v0 | 72±3 | **83±2** | 2±1 | 4±1 | 1±2 | 0±1 | 72±5 | 76±5 | **84±1** | / | / | / | / | / | / | / | **85±7** | / | / | / |
> | visual-antmaze-large-stitch-v0 | 5±5 | 66±16 | 0±0 | 0±0 | 0±1 | 0±1 | / | / | 11±3 | / | 27±3 | / | / | / | / | / | **77±6** | / | / | / |
>
> *Note: SAW (Subgoal Advantage-Weighted Policy Bootstrapping) is another recent baseline we added when replying to Reviewer XeSt.*
>
> TTGS combined with standard agents achieves results competitive with or superior to complex, dedicated planning methods, particularly on the hardest `giant` tasks. We show that simple, modular combinations (Standard Agent + TTGS) can outperform sophisticated monolithic algorithms. This has been previously not known and is in spirit is similar to the contribution of "Rainbow" DQN, showing that the right combination of simple or existing ideas (using a value function for graph search, soft penalty, subgoal selection strategy) yields a new SOTA.
>
> We hope this clarifies that TTGS is not just a reimplementation of SORB, but a method that solves the specific problem of "how to plan with *any* standard value function," a problem that SORB and others solve via specialized training.

---

> > ### Comment · Reviewer_pdK4 · 2025-11-26
> >
> > I am considering your response, and I thought I would provide a small comment in the meantime.
> >
> > - To the best of my knowledge, the "local UVFA" from Huang et al. is simply a critic trained with HER, also trained with no knowledge of planning. I would not see that as specialized training. The fact that several works apply the same high-level planning idea on top of differently trained critics would thus already support the fact that graph search may be widely applicable. I understand that you disagree with this point.
> >
> > - If the previous argument holds, then the key contribution is the soft penalization scheme, which I recognizes as novel and effective. I am considering the significance of this contribution.
> >
> > - I apologize for a mistake in my previous response: I meant to talk about manipulation tasks, not visual. I hope my comment makes more sense. I have no issues with the results you provided in these domains.

---

### Official Review · Reviewer_z4AS · 2025-10-30

**Soundness:** 3
**Presentation:** 3
**Contribution:** 2
**Rating:** 2
**Confidence:** 5

**Summary:**

This paper introduces Test-Time Graph Search (TTGS), a lightweight test-time planning framework that enhances the long-horizon performance of pretrained goal-conditioned reinforcement learning (GCRL) agents. The key idea is to perform graph search using the learned goal-conditioned value function without retraining or fine-tuning the policy. TTGS constructs a weighted state graph from the offline dataset, where edge weights are derived from the frozen value function V(s,g). At inference time, it applies Dijkstra’s shortest-path search on this graph to generate a sequence of intermediate subgoals, which the frozen policy executes sequentially.

**Strengths:**

* The paper presents a training-free test-time planning framework for pretrained goal-conditioned reinforcement learning (GCRL) agents.
While structurally related to prior graph-based methods such as DHRL, NGTE, and GAS—which all learn value or distance functions and perform Dijkstra-based planning—TTGS distinguishes itself conceptually by reusing an already trained goal-conditioned value function from existing offline RL algorithms (e.g., HIQL, GCIQL, QRL) rather than learning new distance estimators or hierarchical policies.

* This reframing of the inference-time usage of a value function as a graph-based planner is a creative reuse of existing components, highlighting that strong long-horizon reasoning can emerge without retraining or additional supervision. In that sense, TTGS’s originality lies not in architectural novelty, but in its minimal yet effective re-interpretation of value-based RL

**Weaknesses:**

### 1. Limited algorithmic novelty beyond existing graph-based GCRL frameworks.

While TTGS is presented as a test-time planning framework, its core structure (value function–based distance estimation, graph construction, and Dijkstra search) is conceptually similar to prior works such as GAS (Baek et al., 2025), NGTE (Park et al., 2024), and DHRL (Lee et al., 2022). These methods learn a value or distance estimator and perform shortest-path search over state graphs to derive subgoal sequences.

The main distinction of TTGS is that it reuses a pretrained goal-conditioned value function rather than learning one jointly with the planner.
However, this difference is largely procedural (when the graph is built) rather than algorithmic. From a systems perspective, both TTGS and GAS follow the same computational pipeline: Value/Distance Learning + Graph Construction + Graph Search.

Therefore, the “training-free” claim appears somewhat overstated. Unless TTGS can demonstrate reuse of a foundation-scale pretrained model across new domains without any retraining or data-specific adaptation, the claimed efficiency advantage remains questionable.

### 2. Insufficient comparison and fairness issues with representation-based offline planners:

The paper highlights its superiority over value-based baselines such as HIQL, GCIQL, and QRL in the main results, but it does not include comparisons with recent temporal-distance representation or graph-based approaches. Although GAS (Baek et al., 2025) is mentioned in the Appendix, a precise and analytical comparison is lacking. The authors claim that constructing the graph by random sampling from the replay buffer yields similar performance to GAS’s proposed node-selection strategy, but this is not convincingly demonstrated.

For example, on antmaze-giant-stitch-v0 and antmaze-large-explore-v0, GAS achieves higher success rates than TTGS.
The paper does not analyze why such differences occur or what aspects of the graph construction contribute to these results.
Moreover, TTGS appears to use per-environment hyperparameter tuning (e.g., τ, T), while GAS results are cited using default hyperparameters from antmaze when reporting performance on humanoidmaze, which is not directly evaluated in GAS.
This asymmetry in evaluation settings undermines the fairness of the comparison.
A proper head-to-head evaluation using identical experimental protocols would make the empirical contribution more convincing.

### 3. Applicability to non-navigational or high-dimensional tasks remains unclear:

Most experiments involve navigation-based maze tasks where geometric distance provides a natural metric. It remains unclear whether TTGS can generalize to non-navigational domains,
such as manipulation tasks where goals are semantic, multimodal, or compositional (e.g., object placement or visual rearrangement). Including non-navigation benchmarks (e.g., numerical and visual manipulations tasks) would help establish broader generality and real-world relevance.

**Questions:**

Please provide your responses to the weaknesses mentioned above.

---

> ### Author Response · Authors · 2025-11-20
> **Official Reply to Reviewer z4AS [1/2]**
>
> We thank the reviewer for their detailed feedback. We appreciate your acknowledgment of our method as a "creative reuse of existing components." Our primary goal was exactly that: to create a practical, lightweight wrapper that unlocks the latent long-horizon capabilities of existing GCRL agents without the complexity of specialized training.
>
> We address your concerns regarding algorithmic novelty, comparison fairness, and non-navigational tasks below.
>
> > ... The “training-free” claim appears somewhat overstated. Unless TTGS can demonstrate reuse of a foundation-scale pretrained model ... the claimed efficiency advantage remains questionable.
>
> We clarify that we use "training-free" to describe the *adaptation workflow* advantage rather than the model scale. In the context of offline RL, this means a practitioner who has already trained and tuned a standard GCRL agent (like HIQL or GCIQL) can apply our method immediately to enable long-horizon capabilities, without further gradient updates, fine-tuning, or complex co-training schemes.
>
> To demonstrate this modularity beyond the baselines in the main text, we applied TTGS to the concurrent state-of-the-art method, SAW (Zhou et al., 2025). As detailed in our response to Reviewer XeSt, TTGS improved SAW's success rate across all tested tasks without any tuning, boosting `humanoidmaze-giant-navigate` from 40% to 79% and `pointmaze-giant-navigate` from 68% to 95%.
>
> The finding that pure test-time planning can significantly improve such a wide range of existing, independently trained agents without requiring any additional training or data is the core contribution we aim to highlight.
>
> > Insufficient comparison and fairness issues with ... GAS ... random sampling ... is not convincingly demonstrated.
>
> We do not claim that GAS's sophisticated node selection is unnecessary for GAS (which must learn its metric from scratch). However, we found that for planning with a *pre-trained* value function, simple random sampling is sufficient. To quantify this, we tested a clustering-based node selection strategy similar to GAS. We are selecting 4000 biggest clusters to make a fair comparison to selecting 4000 random states.
>
> | Environment | TTGS (Random Sampling) | TTGS (Clustering) | Clustering Time (s) |
> | :--- | :--- | :--- | :--- |
> | antmaze-giant-navigate | 67 ± 3 | 69 ± 3 | 706.77 |
> | antmaze-giant-stitch | 70 ± 14 | 69 ± 11 | 1932.88 |
> | antmaze-large-explore | 26 ± 34 | 25 ± 33 | 771.11 |
> | humanoidmaze-giant-navigate | 85 ± 6 |  83 ± 9 | 691.54 |
> | humanoidmaze-giant-stitch | 78 ± 8 | 78 ± 12 | 705.48 |
> | pointmaze-giant-navigate | 73 ± 12 | 74 ± 9 | 814.76 |
> | pointmaze-giant-stitch | 80 ± 6 | 79 ± 10 | 876.43 |
>
> As shown, clustering offers no statistically significant performance advantage for our method, but increases graph construction time from ~40 seconds to >10 minutes. Given this, we believe random sampling is the superior engineering choice for our specific setting (planning with pretrained values). We will include this ablation in the Appendix.
>
> > ...TTGS appears to use per-environment hyperparameter tuning ... while GAS results are cited using default hyperparameters...
>
> This is a valid criticism. To ensure a fair comparison, we re-ran HIQL+TTGS on all locomotion tasks using a **single, fixed set of hyperparameters** derived from `antmaze` ($\tau=24, T=48$). This mirrors the GAS evaluation protocol (using AntMaze settings for all domains).
>
> | Environment | HIQL (Base) | HIQL+TTGS (Fixed Params) | GAS |
> | :--- | :--- | :--- | :--- |
> | antmaze-giant-navigate | 67 ± 4 | 68 ± 3 | 76 ± 6 |
> | antmaze-giant-stitch | 2 ± 1 | 48 ± 19 | 86 ± 4 |
> | antmaze-large-explore | 5 ± 7 | 60 ± 26 | 91 ± 9 |
> | humanoidmaze-giant-navigate | 14 ± 6 | 78 ± 9 | 14 ± 5 |
> | humanoidmaze-giant-stitch | 4 ± 2 | 69 ± 18 | 8 ± 4 |
> | pointmaze-giant-navigate | 47 ± 10 | 68 ± 13 | 5 ± 11 |
> | pointmaze-giant-stitch | 0 ± 0 | 73 ± 14 | 0 ± 0 |
>
> This head-to-head comparison reveals a key trade-off. **GAS achieves higher scores on `antmaze`** (the domain its hyperparameters and representation were tuned for), confirming its strength as a specialized learned planner. However, **TTGS drastically outperforms GAS on `humanoidmaze` and `pointmaze`**, where GAS often achieves near-zero success (e.g., 0% vs 73% on pointmaze-stitch).
>
> This suggests that while GAS may have a higher performance ceiling when tuned/trained for a specific environment, TTGS is significantly more robust across diverse domains despite using fixed hyperparameters. We will update the paper to include this direct comparison table to transparently show where each method excels.

---

> ### Author Response · Authors · 2025-11-20
> **Official Reply to Reviewer z4AS [2/2]**
>
> > Applicability to non-navigational ... tasks remains unclear.
>
> We expanded our evaluation to manipulation tasks from OGBench, pairing TTGS with GCIQL (the strongest base agent there).
>
> | Environment | GCIQL Success | GCIQL+TTGS Success |
> | :--- | :--- | :--- |
> | scene-play | 50 ± 7 | 52 ± 4 |
> | cube-triple-play | 4 ± 2 | 4 ± 2 |
> | puzzle-4x5-play | 12 ± 3 | 13 ± 3 |
> | puzzle-4x6-play | 10 ± 0 | 10 ± 0 |
>
> The gains are consistent but modest. To understand why, we analyzed the geometry of the value function by plotting dataset states on a 2D plane where $x$ is the predicted distance from the start and $y$ is the predicted distance to the goal.
> *   **Locomotion:** States typically form a curve connecting the start and goal (low $x+y$), offering a dense set of intermediate subgoals.
> *   **Manipulation:** Dataset states cluster far from both start and goal (high $x$ and high $y$), with very few states bridging the gap between start and goal.
>
> This analysis indicates that for these manipulation datasets, the evaluation goals often lie far from the data manifold. Crucially, TTGS correctly identifies that no reliable path exists through the graph (due to high edge costs across the gap) and defaults to the base policy behavior. This confirms that TTGS does not hallucinate plans when the data support is missing. We will add these plots and analysis to the paper to characterize the method's behavior on unconnected data manifolds.
>
> We hope this direct comparison and the new experiments address your concerns regarding fairness and scope. Is there other information we can provide that would help the reviewer in raising their score?

---

### Official Review · Reviewer_XeSt · 2025-10-31

**Soundness:** 4
**Presentation:** 4
**Contribution:** 3
**Rating:** 6
**Confidence:** 4

**Summary:**

The work studies goal-conditioned reinforcement learning (GCRL) on long-horizon tasks.
The work observes that the policies trained by common GCRL algorithms do not exploit all geometric structure learned by the goal-conditioned value function.
They propose a training-free test-time method that leverages only the offline pre-training data to improve policies during evaluation.
Their proposed method constructs a directed graph of subsampled states, leveraging the goal-conditioned value function to compute distances.
At the beginning of an evaluation episode, Dijkstra's algorithm is used to find a shortest path to the goal, and during evaluation the agent adaptively attempts to reach one of the subgoals along the estimated shortest path.

**Strengths:**

The paper proposes a novel method for GCRL and extensively benchmarks it against several common GCRL algorithms on hard tasks.
The experimental results in long-horizon planning are strong and convincing.
Further, the paper includes extensive ablation studies.

**Weaknesses:**

* Results would be stronger if compared against more recent "hierarchy-inspired" methods such as SAW [1] or test-time methods such as GC-TTT [2].
* The proposed method is not truly adaptive during evaluation, since the determined shortest path remains fixed. It may be beneficial to recompute a shortest path if the agent goes astray from the originally computed path at intermediate subgoals.
* Building an explicit graph (even if only over subsampled states) may be prohibitive in larger environments.
* Based on the results of the paper, a natural question is whether the proposed explicit hierarchical search could be distilled into a high-level policy (similar to HIQL). In other words, are there specific design choices in HIQL / the architecture that prevent effective training of the high-level policy?
* The optimal hyperparameters seem to depend relatively strongly on the environment (though the performance gains over the initial policy are robust).

[1]: Zhou et al., Flattening Hierarchies with Policy Bootstrapping. https://arxiv.org/pdf/2505.14975
[2]: Bagatella et al., Test-time Offline Reinforcement Learning on Goal-related Experience. https://arxiv.org/pdf/2507.18809

**Questions:**

* In figure 2, what is your understanding for why HIQL fails in this example?
* Did you ablate the clipping of $\hat{d}$?
* Can you comment on the relation to other test-time methods for GCRL such as GC-TTT [2, above]?

---

> ### Author Response · Authors · 2025-11-20
> **Official Reply to Reviewer XeSt [1/2]**
>
> We appreciate your positive assessment of our work’s soundness and presentation. We are grateful for your constructive feedback regarding concurrent work and adaptivity, which has motivated us to conduct additional experiments to further strengthen our contribution.
>
> > Results would be stronger if compared against more recent "hierarchy-inspired" methods such as SAW [1] or test-time methods such as GC-TTT [2].
>
> We agree that positioning TTGS relative to these concurrent works (both released May/July 2025) is important. We view TTGS as **complementary** to methods like SAW and GC-TTT. While those methods aim to improve the base policy or fine-tune it, TTGS is a planning wrapper that can act on top of *any* policy.
>
> To demonstrate this, we integrated TTGS with the publicly available SAW policy using our default hyperparameters ($\tau=24, T=48, M=4000$) without any tuning. As shown below, TTGS consistently boosts SAW's performance, confirming that our method can extract further gains even from state-of-the-art policies.
>
> | Environment | SAW (Base) Success  | SAW + TTGS Success |
> | :--- | :--- | :--- |
> | antmaze-giant-navigate | 69 ± 3 | 71 ± 5 |
> | humanoidmaze-giant-navigate | 40 ± 3 | 79 ± 5 |
> | pointmaze-giant-navigate | 68 ± 6 | 95 ± 3 |
> | visual-antmaze-giant-navigate | 4 ± 1 | 8 ± 4 |
> | visual-antmaze-large-navigate | 72 ± 5 | 76 ± 5 |
>
> GC-TTT operates by fine-tuning the policy via gradient updates on retrieved data during execution. In contrast, TTGS acts as a lightweight planner over frozen modules. These are distinct paradigms: one optimizes the controller, the other optimizes the route. We believe these approaches could be combined: GC-TTT can be used to adapt the local policy while TTGS provides the high-level waypoints. We will update the related work section to clarify these distinctions and include the SAW results to highlight modularity.
>
> > The proposed method is not truly adaptive during evaluation, since the determined shortest path remains fixed. It may be beneficial to recompute...
>
> This is an excellent point. While the *guide path* is fixed, the *subgoal selection* is fully adaptive. As detailed in Algorithm 1, at every step the agent identifies the closest waypoint on the guide path to its *current* state and selects a reachable subgoal *ahead* of it. This allows continuous self-correction toward the global plan without the cost of full replanning.
>
> To test if explicit replanning is needed, we ran an ablation (also shared with Reviewer M9q6) where we re-run Dijkstra if the agent strays $>2T$ from the current subgoal.
>
> | Environment | HIQL (Base) | TTGS (No Replan) | TTGS (With Replan) | TTGS Replan Episode Ratio |
> | :--- | :--- | :--- | :--- | :--- |
> | antmaze-giant-navigate | 66 ± 3 | 67 ± 3 | 71 ± 2 | 0.52 |
> | antmaze-giant-stitch | 1 ± 1 | 70 ± 14 | 67 ± 13 | 0.38 |
> | antmaze-large-explore | 2 ± 3 | 26 ± 34 | 50 ± 32 | 0.07 |
> | humanoidmaze-giant-navigate | 14 ± 5 | 85 ± 6 | 78 ± 4 | 0.52 |
> | humanoidmaze-giant-stitch | 4 ± 3 | 78 ± 8 | 33 ± 10 | 0.98 |
> | pointmaze-giant-navigate | 50 ± 13 | 73 ± 12 | 72 ± 6 | 0.01 |
> | pointmaze-giant-stitch | 0 ± 0 | 80 ± 6 | 80 ± 6 | 0.0 |
> | visual-antmaze-large-navigate | 46 ± 14 | 82 ± 7 | 70 ± 10 | 0.86 |
> | visual-antmaze-large-stitch | 18 ± 8 | 78 ± 4 | 53 ± 24 | 0.76 |
>
> Replanning significantly helps in `antmaze-large-explore`, where the base policy is unreliable and frequently deviates. However, in most tasks, it offers no benefit or even degrades performance (likely due to the agent oscillating between two competing optimal paths). This confirms that for competent base policies, our adaptive subgoal selection is sufficient. We will include this result and a discussion on when replanning might be beneficial in the appendix.
>
> > Building an explicit graph (even if only over subsampled states) may be prohibitive in larger environments.
>
> We acknowledge this scalability constraint in our Limitations section. However, we emphasize that OGBench `giant` tasks are currently the largest standard benchmarks available for offline GCRL. For these tasks, our random sampling approach ($M=4000$) takes under 100 seconds for graph construction and is highly effective. We believe exploring different methods for graph construction is a promising direction for future work to scale beyond current benchmarks.
>
> > ...whether the proposed explicit hierarchical search could be distilled into a high-level policy...
>
> We completely agree. The globally consistent paths found by TTGS would likely provide a much higher quality training signal than the single-trajectory future states used by HIQL. We view TTGS as establishing the value of explicit planning, with distillation being a natural next step.

---

> ### Author Response · Authors · 2025-11-20
> **Official Reply to Reviewer XeSt [2/2]**
>
> > In figure 2, what is your understanding for why HIQL fails in this example?
>
> HIQL's learned high-level policy likely suffers from long-horizon credit assignment issues, predicting subgoals that are locally valid but do not form a globally connected path to the final goal. TTGS solves this by enforcing connectivity via explicit graph search.
>
> > Did you ablate the clipping of d?
>
> We clip the minimum distance to 1 because at least one action is required to change states. We also must clip negative predictions because Dijkstra's algorithm mathematically requires non-negative edge weights. We view these as structural validity constraints rather than hyperparameters. If you feel an ablation is critical, could you please suggest specific clipping values you would be interested in?
>
> We hope the new results with SAW and the replanning ablation strengthen your confidence in our work. Is there any additional information we can provide that might encourage you to raise your score?

---

### Official Review · Reviewer_M9q6 · 2025-10-31

**Soundness:** 2
**Presentation:** 2
**Contribution:** 2
**Rating:** 2
**Confidence:** 5

**Summary:**

Offline GCRL methods struggle especially in long-horizon tasks because of temporal credit assignment problem. Recent approaches propose decomposing problem into reachable subproblems via hierarchical methods or graph-search methods, but they still require specialized training or additional data beyond the offline training data. This paper proposes a test-time adaptation approach for improving the performance of offline GCRL without requiring additional data during the test time. The idea is first building a graph of distances (calculated using the learned value function) using samples from the offline dataset, then finding the shortest path from start to the goal, and selecting reachable subgoals on the path. Experimental results shows graph-based test time fine-tuning improves the performance.

**Strengths:**

**Interesting Approach:** Using test-time fine tuning for offline GCRL via constructing a graph is interesting.

**Improved Performance:** Experimental results shows performance improvements over baselines.

**Weaknesses:**

- **Computational Complexity**: The proposed approach requires constructing a graph based on large offline datasets, then calculating the shortest path, and selecting subgoals at test deployment. Because of the graph construction and shortest-path calculation, the proposed method seems computationally complex. Therefore, the computational overhead over baselines must be discussed clearly. Wall-clock times and FLOPs should be presented for both TTGS and baselines to show how much computational overhead is required.

- **Following the Shortest Path**: The idea is built on the assumption that the agent will initially follow the shortest path, since the shortest path is only calculated once. If the agent deviates from this path, it might become unusable. This must be discussed clearly, because due to the credit assignment problem in offline RL, the value function is generally noisy, which might lead the agent away from the shortest path. What happens if the agent ends up in a different corner, far from the shortest path? Have you ever considered re-calculating the shortest path at regular intervals during test time?

- **Distance Prediction**: It is stated that the authors find the value function "very reliable" for calculating the distance near goal states, which is not clearly presented or discussed.

- **Experimental Setting**: The proposed approach is only evaluated on five long-horizon tasks against baselines. The applicability of the proposed approach to short- and medium-horizon tasks should be discussed, and if possible, those tasks should be included in experiments. In addition, some environments in Table 1 are omitted from Figure 3, which is concerning. Please report results for all environments in Table 1 in Figure 3 as well.

**Questions:**

- How computationally complex is the proposed approach?
- Why is the shortest path only calculated once at the beginning of the test? Have you ever considered re-calculating the shortest path from the current state to the goal at certain intervals during test time deployment?
- What happens if the agent deviates from the calculated shortest path because of the noisy value function it learned during training? Since the shortest path is calculated once, when the agent deviates from it, does this render this shortest path sub-optimal?
- Why are some environments in Table 1 not reported in Figure 3?
- Can you please also elaborate on the applicability of the proposed approach to short- and medium-horizon tasks?

---

> ### Author Response · Authors · 2025-11-20
> **Official Reply to Reviewer M9q6 [1/2]**
>
> We thank the reviewer for their time and feedback. We are glad you found our test-time graph search approach interesting. We address your concerns regarding computational complexity, robustness to deviation, and experimental scope below.
>
> > Computational Complexity: ... Because of the graph construction and shortest-path calculation, the proposed method seems computationally complex. ... Wall-clock times and FLOPs should be presented...
>
> While we provided general timing estimates in the Limitations section (lines 439-445), we agree that a detailed breakdown is more rigorous. The graph construction is performed once per dataset, and the shortest path search is performed once per episode (or sparingly if replanning). Both operations are matrix-heavy and highly parallelizable; we implement them on the GPU, where the $O(M^2)$ complexity for pairwise distances and Dijkstra is handled efficiently.
>
> We present the wall-clock times below (measured on an Nvidia L40s GPU). Graph construction takes $\approx 36$s for state-based tasks and $\approx 100$s for visual tasks. The per-episode planning overhead is $<1$s. The per-step action selection overhead is negligible ($\approx 1.5$ ms).
>
> | Environment | Graph Build (s) | Shortest Path Search (s) | TTGS Step Overhead (ms) |
> | :--- | :--- | :--- | :--- |
> | antmaze-giant-navigate | 36.55 | 0.46  | 1.42 |
> | antmaze-giant-stitch | 36.26 | 0.84 | 1.66 |
> | antmaze-large-explore | 36.40 | 0.46 | 1.33 |
> | humanoidmaze-giant-navigate | 34.99 | 0.59 | 1.39 |
> | humanoidmaze-giant-stitch | 36.35 | 1.0 | 1.38 |
> | pointmaze-giant-navigate | 36.42 | 1.25  | 1.68 |
> | pointmaze-giant-stitch | 34.98 | 0.79  | 1.61 |
> | visual-antmaze-large-navigate | 100.02 | 0.53  | 2.63 |
> | visual-antmaze-large-stitch | 100.89 | 0.53  | 2.49 |
>
> We will include this detailed table in the Appendix. Does this breakdown alleviate your concerns regarding the computational feasibility of our method?
>
> > Following the Shortest Path: The idea is built on the assumption that the agent will initially follow the shortest path... If the agent deviates from this path, it might become unusable. ... Have you ever considered re-calculating the shortest path...?
>
> Our algorithm is explicitly designed to handle deviation without requiring constant full replanning. As described in Section 3.3 (Adaptive Subgoal Selection), the pre-computed path acts as a *guide*, not a rigid trajectory. At every step, the agent identifies the closest waypoint on the guide to its *current* state and selects a reachable subgoal ahead of it. This allows the agent to continuously self-correct locally.
> Qualitatively, we observe that failures are rarely caused by the agent "getting lost" (deviating far from the path); rather, they are caused by the base policy getting stuck at specific bottlenecks.
>
> However, to empirically test your hypothesis, we ran a new ablation where we trigger a full recalculation of the shortest path if the agent deviates from the current subgoal by more than $2T$ (at most every 50 steps to avoid excessive path oscillations).
>
> | Environment | HIQL (Base) | TTGS (No Replan) | TTGS (With Replan) | Episodes With Replanning Ratio |
> | :--- | :--- | :--- | :--- | :--- |
> | antmaze-giant-navigate | 66 ± 3 | 67 ± 3 | 71 ± 2 | 0.52 |
> | antmaze-giant-stitch | 1 ± 1 | 70 ± 14 | 67 ± 13 | 0.38 |
> | antmaze-large-explore | 2 ± 3 | 26 ± 34 | *50 ± 32* | 0.07 |
> | humanoidmaze-giant-navigate | 14 ± 5 | 85 ± 6 | 78 ± 4 | 0.52 |
> | humanoidmaze-giant-stitch | 4 ± 3 |* 78 ± 8* | 33 ± 10 | 0.98 |
> | pointmaze-giant-navigate | 50 ± 13 | 73 ± 12 | 72 ± 6 | 0.01 |
> | pointmaze-giant-stitch | 0 ± 0 | 80 ± 6 | 80 ± 6 | 0.0 |
> | visual-antmaze-large-navigate | 46 ± 14 | 82 ± 7 | 70 ± 10 | 0.86 |
> | visual-antmaze-large-stitch | 18 ± 8 | 78 ± 4 | 53 ± 24 | 0.76 |
>
> In `antmaze-large-explore` replanning indeed helps significantly ($26\\% \to 50\\%$). We hypothesize that this is due to the base policy being unreliable since it was trained on noisy exploratory data, leading to significant deviations. However, across most environments, frequent replanning offers no benefit and can sometimes degrade performance. We observed that frequent replanning can cause the agent to oscillate between two equally optimal paths or subgoals, hindering forward progress. We will incorporate this ablation into the paper to clarify that while our method supports replanning, the one-time adaptive guide is sufficient for most tasks.

---

> ### Author Response · Authors · 2025-11-20
> **Official Reply to Reviewer M9q6 [2/2]**
>
> > Experimental Setting: The proposed approach is only evaluated on five long-horizon tasks ... some environments in Table 1 are omitted from Figure 3 ... Please report results for all environments in Table 1 in Figure 3 as well.
>
> Figure 3 was intended only to highlight the hardest `giant-stitch` tasks. The full results for all 15 environments (including `navigate`, `stitch`, `explore`, and visual tasks) were included in **Table 3 in Appendix A**. We will revise the main text to reference Appendix A more prominently so these results are not missed.
>
> Regarding short- and medium-horizon tasks: Our primary focus is long-horizon failures, but we agree that verifying robustness on shorter tasks is important. Since OGBench does not include "small" tasks, we evaluated TTGS on the `medium` and `large` configurations. We used ($\tau=24, T=48, M=4000$) without any tuning.
>
> | Environment | HIQL Success | HIQL+TTGS Success |
> | :--- | :--- | :--- |
> | antmaze-medium-navigate | 95 ± 1 | 95 ± 1 |
> | antmaze-medium-stitch | 93 ± 2 | 95 ± 1 |
> | antmaze-large-navigate | 91 ± 3 | 92 ± 2 |
> | antmaze-large-stitch | 73 ± 6 | 91 ± 2 |
> | humanoidmaze-medium-navigate | 88 ± 3 | 93 ± 3 |
> | humanoidmaze-medium-stitch | 85 ± 3 | 86 ± 3 |
> | humanoidmaze-large-navigate | 48 ± 5 | 79 ± 6 |
> | humanoidmaze-large-stitch | 29 ± 3 | 65 ± 11 |
> | pointmaze-medium-navigate | 74 ± 4 | 86 ± 5 |
> | pointmaze-medium-stitch | 73 ± 10 | 80 ± 20 |
> | pointmaze-large-navigate | 45 ± 14 | 78 ± 10 |
> | pointmaze-large-stitch | 13 ± 8 | 92 ± 5 |
>
> TTGS consistently maintains or improves performance on these tasks, confirming it does not degrade behavior when the horizon is shorter. We will add these results to the comprehensive table in the Appendix.
>
> > Distance Prediction: It is stated that the authors find the value function "very reliable" for calculating the distance near goal states, which is not clearly presented or discussed.
>
> By "reliable," we mean that the value function accurately reflects *local* reachability (e.g., within a short horizon), even if its global long-horizon estimates are noisy. The strong empirical performance of TTGS (which relies on chaining these local estimates) and the visual coherence of the distance fields in Figure 4b serve as evidence of this local reliability. We will clarify this distinction in the text.
>
> We hope these additional experiments and clarifications address your concerns. Given the new evidence of computational efficiency and robustness, would you be willing to reconsider your score? Can we provide any additional information to help with this decision?

---

### Author Response · Authors · 2025-12-04
**Summary of Revisions and Author Response**

During the rebuttal period, we extensively revised the manuscript and conducted new experiments to address all reviewer feedback. Because the visible reviews and scores reflect the submission prior to these major updates, we provide this summary to detail the changes. We highlight below how we have resolved the primary concerns regarding comparison fairness, computational overhead, and algorithmic novelty. We believe the revised paper and new empirical evidence warrant a re-evaluation of the work.

### 1. Summary of Revisions & New Experiments
During the rebuttal, we updated the manuscript with the following key additions:
*   **Integration with SAW:** We integrated TTGS with **SAW** (Zhou et al., 2025), a concurrent state-of-the-art method. TTGS improved SAW’s performance across the vast majority of tested tasks without any tuning (e.g., `humanoidmaze-giant-stitch` from 0% $\to$ 79.8%), demonstrating our method's modularity and effectiveness.
*   **Head-to-Head Comparison with GAS:** We added a comparison with GAS (Baek et al., 2025) using fixed hyperparameters. We show that while GAS excels in the specific domain it was tuned for, TTGS is significantly more robust, outperforming GAS on `pointmaze` (**73% vs 0%**) and `humanoidmaze` (**69% vs 8%**) when using fixed hyperparameters (Table 4).
*   **Soft vs. Hard Penalty Ablation:** We added an ablation (Figure 4(a), Table 7) demonstrating that our soft penalty is essential. A standard hard-threshold approach (common in prior graph methods) fails on difficult stitching tasks, proving the necessity of our specific design choice.
*   **Runtime Analysis:** We added a detailed breakdown (Table 8) showing graph construction takes only ~36s per dataset and inference overhead is negligible (<1.5ms per step), addressing computational concerns.
*   **Replanning Ablation:** We analyzed online replanning (Table 9) and found that our adaptive subgoal selection on a fixed guide path is generally sufficient and often more stable than frequent replanning.
*   **Manipulation Tasks:** We expanded evaluation to manipulation domains (Table 5) and analyzed value function geometry (Figure 5) to explain behavior on unconnected manifolds.

---

> ### Author Response · Authors · 2025-12-04
>
> ### 2. Addressing Specific Reviewer Concerns
>
> **To Reviewer M9q6 (Score: 2)**
> *   **Concern:** The reviewer expressed concern regarding the evaluation scope.
> *   **Response:** We believe this assessment overlooked **Table 3 in Appendix A**, which contains full results for 27 environments, including the medium/large horizon tasks added during rebuttals. We have made the reference to these results more prominent in the main text.
> *   **Concern:** Computational complexity and value reliability.
> *   **Response:** We added **Table 8**, proving the method is highly efficient. We also clarified that "reliability" refers to *local* reachability; our **soft penalty** mechanism successfully chains these local estimates into reliable global plans, as evidenced by the high success rates.
>
> **To Reviewer z4AS (Score: 2) & pdK4 (Score: 2)**
> *   **Concern:** Novelty compared to SORB/GAS and the perception of TTGS as standard graph search.
> *   **Response on Novelty:** Prior methods like SORB require **specialized training** (Distributional RL, ensembles, specific losses, or data selection) to make graph search work. TTGS is a **training-free wrapper** that unlocks planning capabilities in *standard, off-the-shelf* agents (HIQL, GCIQL, QRL, SAW).
>     *  We demonstrate empirically (Table 7) that applying standard graph search (Hard Threshold) to these agents **fails** (e.g., success drops from 70% to 2% on `antmaze-giant-stitch`), proving that naive application of graph search is insufficient.
>     *  Our contribution is identifying the specific mechanisms (Soft Penalty, Subgoal Selection) that allow standard agents to support planning without retraining. This is a novel finding that makes graph search accessible to standard GCRL agents. The magnitude of improvement achievable by standard agents via graph search is also a key new result.
>     *  We updated the main text to better highlight the novel contributions of our work and differentiate it from related methods.
> *   **Response on Fairness (z4AS):** We addressed the comparison with GAS by running a new experiment with identical, fixed hyperparameters (Table 4). This revealed that TTGS is actually *more* robust across diverse domains than the learned planner in GAS.
> *  **Clustering ablation**: We included a new experiment (Table 6) confirming that clustering states for graph vertices does not provide significant benefits over random sampling for our method, while significantly increasing build time.
>
> **To Reviewer XeSt (Score: 6)**
> *   **Concern:** Comparison to recent hierarchy/test-time methods and replanning.
> *   **Response:** We successfully combined TTGS with SAW (a very recent hierarchy method), showing additive gains. We also clarified that TTGS is complementary to test-time fine-tuning methods like GC-TTT. Our replanning ablation (Table 9) confirmed that our adaptive subgoal selection is robust, addressing the reviewer's query about deviation handling.
>
> ### 3. Conclusion
> This paper presents a practical, "plug-and-play" solution that significantly improves long-horizon capabilities of standard Offline GCRL agents. By demonstrating how to utilize the latent geometric structure in standard value functions via a novel soft-penalty mechanism in combination with adaptive subgoal selection, we achieve state-of-the-art performance without the complexity of specialized training.
>
> We respectfully ask the Area Chair to consider these revisions and the strong empirical evidence provided (particularly the new comparisons with SAW and GAS) when making a decision.

---

### Meta-Review · Area_Chair_nQXe · 2026-01-07

**Summary:**

The primary concerns that weighed against acceptance were:
1. Novelty and relationship to prior work (pdK4, z4AS)
* SORB equivalence: pdK4 argued TTGS is methodologically identical to SORB's inference wrapper, making the "training-free" distinction procedural rather than algorithmic. The soft penalty was seen as the only novel component, and its necessity was questioned.
* Limited algorithmic innovation: z4AS viewed TTGS as structurally similar to GAS, NGTE, and DHRL, differing mainly in when the graph is built rather than how.
2. Computational and scalability issues (M9q6, XeSt)
* Overhead: M9q6 requested detailed complexity analysis (wall-clock, FLOPs) and criticized the single-path planning approach as brittle.
* Scalability: XeSt questioned whether explicit graph construction would be prohibitive beyond current benchmarks.
3. Robustness and adaptivity (M9q6, XeSt)
* Deviation handling: M9q6 challenged the assumption that agents follow the precomputed shortest path, especially given noisy value functions.
Fixed path limitation: XeSt noted the method isn't "truly adaptive" since the shortest path is computed once.
4. Empirical breadth and fairness (M9q6, z4AS)
* Limited evaluation scope: M9q6 criticized evaluation on only five long-horizon tasks and missing results in Figure 3.
* Comparison fairness: z4AS pointed out asymmetric hyperparameter tuning (TTGS per-environment vs. GAS defaults) and lack of direct head-to-head comparison.
* Domain generalization: Both z4AS and pdK4 questioned applicability beyond navigation tasks like manipulation.
5. Hyperparameter sensitivity (XeSt)
* Concern that optimal hyperparameters ($\tau$, T) vary significantly across environments.

**Reviewer Concerns:**

Effectively addressed
* Computational  complexity: Authors provided comprehensive timing data (Table 8) showing graph construction ~36s and per-step overhead <1.5ms, demonstrating feasibility. M9q6's concern was directly resolved with empirical evidence.
* Replanning/deviation robustness: The replanning ablation (Table 9) convincingly showed that adaptive subgoal selection handles deviations well, and frequent replanning is often unnecessary or harmful. This satisfied both M9q6 and XeSt.
* Comparison with  concurrent methods:
* SAW integration: New experiments showed TTGS improves SAW across tasks, demonstrating modularity and complementarity to hierarchy methods (addressing XeSt).
* GAS fairness: Head-to-head comparison with fixed hyperparameters revealed TTGS is more robust across diverse domains (humanoidmaze, pointmaze) while GAS excels only on its tuned domain (antmaze). This directly addressed z4AS's fairness critique.
* Soft penalty necessity: The hard vs. soft penalty ablation (Table 7) provided strong evidence that soft penalty is essential for challenging "stitch" tasks (performance drops from ~70% to ~2% on antmaze-giant-stitch), validating the core design choice for pdK4 and z4AS.
* Empirical scope: Added medium/large task results and promised prominent Appendix referencing, addressing M9q6's scope concern.
* Node selection strategy: Clustering vs. random sampling ablation showed random sampling is superior for TTGS, justifying the design and responding to z4AS.
* Manipulation tasks: While improvements were modest, the evaluation showed TTGS doesn't hallucinate plans when data support is missing, demonstrating robustness and partially addressing domain generalization concerns.

Remaining:
* Novelty vs. SORB: The fundamental disagreement with pdK4 persists. pdK4 maintains that SORB's inference wrapper is equally training-agnostic and that subsequent works already demonstrated standard value functions support graph search. Authors' distinction about "specialized training requirements" remains contested, as pdK4 argues distributional RL and ensembles are SORB's training contributions, not inference requirements.
* Soft penalty "essential" claim: pdK4 argues the term is overstated, noting hard thresholds work "nearly as well" on some domains and could be computed dynamically to avoid disconnections. The static soft penalty's superiority isn't universally convincing.
* Manipulation task: The modest gains (e.g., cube-triple-play: 4% → 4%) leave open whether TTGS provides meaningful benefits in less geometrically-structured domains, a core concern for z4AS and pdK4.
* Scalability beyond current benchmarks: While authors acknowledged the limitation, no concrete solutions or theoretical analysis were provided for scaling to truly massive state spaces (XeSt's concern remains valid).
* Hyperparameter sensitivity: Though authors showed robustness with fixed parameters across locomotion tasks, the broader claim of insensitivity wasn't fully substantiated across all domains (XeSt's concern partially remains).
* Distillation potential: XeSt's question about why HIQL fails and whether search could be distilled into a learned high-level policy was largely deferred as "future work" rather than directly addressed.

**Reviewer Scores:**

Reviewer M9q6: This reviewer was highly confident (5/5) but their concerns were empirical and well-defined. The rebuttal directly addressed major issue with new data. While the reviewer would appreciate this rigorous response, their initial "absolutely certain" rejection stance suggests they'd move cautiously to a borderline score rather than full acceptance.

Reviewer z4AS: Despite strong rebuttal evidence, this reviewer's high confidence (5/5) in the novelty critique would make the reviewer hesitate to fully endorse acceptance

Reviewer pdK4: This reviewer explicitly stated they would keep the score after the rebuttal, indicating fundamental disagreement remains. I think the reviewer would likely remain at reject, possibly moving to 4 if convinced by the soft penalty ablation's dramatic stitch-task failures.

---

### Decision · Program_Chairs · 2026-01-26

Reject